# On the origin of elasticity and heat conduction anisotropy of liquid crystal elastomers at gigahertz frequencies

Yu Cang[1,2], Jiaqi Liu [ID][3], Meguya Ryu[4,5], Bartlomiej Graczykowski [ID][2,6], Junko Morikawa[4], Shu Yang [ID][3] ✉ & George Fytas [ID][2] ✉

Liquid crystal elastomers that offer exceptional load-deformation response at low frequencies often require consideration of the mechanical anisotropy only along the two symmetry directions. However, emerging applications operating at high frequencies require all five true elastic constants. Here, we utilize Brillouin light spectroscopy to obtain the engineering moduli and probe the strain dependence of the elasticity anisotropy at gigahertz frequencies. The Young's modulus anisotropy, $E_{\parallel}/E_{\perp} \sim 2.6$, is unexpectedly lower than that measured by tensile testing, suggesting disparity between the local mesogenic orientation and the larger scale orientation of the network strands. Unprecedented is the robustness of $E_{\parallel}/E_{\perp}$ to uniaxial load that it does not comply with continuously transformable director orientation observed in the tensile testing. Likewise, the heat conductivity is directional, $\kappa_{\parallel}/\kappa_{\perp} \sim 3.0$ with $\kappa_{\perp} = 0.16\,\mathrm{Wm^{-1}K^{-1}}$. Conceptually, this work reveals the different length scales involved in the thermoelastic anisotropy and provides insights for programming liquid crystal elastomers on-demand for high-frequency applications.

Elastic anisotropy is a remarkable feature of liquid crystal elastomers (LCEs) inherited from the assembly of the elongated liquid crystal (LC) mesogens, the units in the polymer networks, either on the main chain or on the side-chain[1,2]. When the mesogens are aligned in a certain direction during the LCE fabrication, the polymer chains adopt an elongated conformation in an affine way, conferring an elastic modulus a few times (2–10) larger along the longitudinal (∥) direction than that of normal (⊥) to it[3–6]. Therefore, when heated above the nematic to isotropic phase transition temperature ($T_{\mathrm{NI}}$), the network will shrink in the aligned direction while expanding in the perpendicular direction with near zero stress[7,8]. Programming LCE director fields has shown great promises for applications, including soft robotics and actuators[9–12], origami[7,13], and shape transformation from two dimensions (2D) to three dimensions with arbitrary curvatures[14] and inflatable artificial chromatophores capable of broadband color-shifting[15]. For such applications, Young's modulus $E$ and its anisotropy ($E_{\parallel}/E_{\perp}$), typically measured in tensile testing at low frequencies[3–6,9,16], represent the static mechanical properties of LCEs with glass transition temperature below the ambient temperature. However, for high-frequency applications such as radio-frequency (RF) packaging[17], 5G networks[18], and self-morphing antennas[19] with operating frequencies in the gigahertz (GHz) range, the elastic response of LCE cannot be described by the conventional macroscopic mechanics characterized by tensile testing. Instead, access to the high-frequency $E$ and $E_{\parallel}/E_{\perp}$ is necessary but missing in the literature.

[1]School of Aerospace Engineering and Applied Mechanics, Tongji University, Zhangwu Road 100, Shanghai 200092, China. [2]Max Planck Institute for Polymer Research, Ackermannweg 10, Mainz 55128, Germany. [3]Department of Materials Science and Engineering, University of Pennsylvania, 3231 Walnut Street, Philadelphia, PA 19104, USA. [4]School of Materials and Chemical Technology, Tokyo Institute of Technology, Ookayama, Meguro-ku, Tokyo 152-8550, Japan. [5]National Metrology Institute of Japan (NMIJ), National Institute of Advanced Industrial Science and Technology (AIST), Umezono, Tsukuba 305-8563, Japan. [6]Faculty of Physics, Adam Mickiewicz University, Uniwersytetu Poznanskiego 2, Poznan 61-614, Poland. ✉e-mail: shuyang@seas.upenn.edu; fytas@mpip-mainz.mpg.de

Fundamentally, the orientational contributions of the mesogens and the network polymer strands to the mechanical anisotropy can be frequency-dependent due to their very different dynamics. For small molecule LCs, such as n-alkyl cyanobiphenyl (n-CB, $n = 5$–$9$)[20] and phenyl pyrimidine[21] LCs, the ratio of the longitudinal modulus in the parallel vs. orthogonal directions measured at GHz frequencies is typically <1.15[21]; in the absence of polymer network this anisotropy reflects contributions solely by the mesogens. For LCE $E_{\parallel}/E_{\perp}$ from low-frequency tensile testing is much larger (-3.3)[22] and is considered as the static anisotropy value due to the contribution of both aligned network and mesogens. Different from small molecule LCs, LCEs uniquely feature both local mesogenic and network strand structures, so that access to the mechanical properties at GHz frequencies will elucidate the origin of enhanced elasticity anisotropy in LCEs and response to deformation at the microscopic level[23]. Yet, it raises an important but so far neglected, fundamental question in the classic LCE network investigation, that is how the molecular ordering or alignment of mesogenic units are coupled within the polymer network, thus, impacting their thermal and mechanical responses at different frequencies. When probed at low frequencies, the LCE is in the elastomeric state where the network polymer strands are mobile, whereas they become frozen when probed at GHz, and thus the dependence on the orientation order parameter must be different.

For electronic and communication device applications[18,19], heat management within the polymer networks is also crucial. As the operating frequency increases, both the power density and the heat density increase continuously, threatening the device reliability[24]. In view of the low thermal conductivity of conventional isotropic polymers, how to improve heat dissipation via the design of polymer chain topology becomes an emergent issue. An efficient strategy is to align the polymer chains, e.g., by uniaxially stretching the polymers, such that we can boost the heat transport along the chains[25–27]. So far, there is limited literature on the anisotropy of thermal conductivity in LCEs with scattered data. Since material elasticity and anisotropy are strongly dependent on chemical compositions and preparation methods, the reported thermal conductivity anisotropy shows huge variations (1.5–15)[28–30] without an obvious correlation to the mean orientation order parameter. The anisotropic thermal conductivity of LCEs is not understood due to the limited amount of data and also the experimental methodology is not well-defined[31]. Therefore, quantitative measurements of the sound velocity and heat conductivity anisotropy from the same sample can elucidate the role of the phonon-mean-free path anisotropy in the directional transport of LCE.

The mechanical properties and their anisotropy in LCEs are conventionally characterized by macroscopic techniques such as tensile testing[14,22,32,33] and dynamic mechanical analysis[34–37]. These methods can access one or two elastic constants, while the complete elastic tensor, containing five independent elastic constants for LCEs, requires several independent experimental arrangements by neglecting the director rotations under strain[38]. Besides, the mechanical response of LCE exhibits a strong dependence on probed temperature and frequency due to its viscoelastic feature[34–37], consequently encumbering access to the purely elastic $E_{\parallel}$ of LCE (in the rubbery state) by conventional characterizations. Elastic material response ($E_{\parallel,\perp}$ typically in the GPa range) is ensured for sufficiently high frequency (compared to the fastest material dynamics) applied isothermally and not artificially by time-temperature (t-T) superposition of low-frequency scan tests at different temperatures[34–37]. Therefore, there remains a limited understanding of LCE mechanical anisotropy, especially on the microscopic scale. Here, we use Brillouin light spectroscopy (BLS), which offers a unique and the only not contact tool for evaluating the elasticity[39] of LCEs at GHz frequencies through the measurement of the sound velocities and attenuation of acoustic phonons at different wavelengths and polarizations (Fig. 1d). At such a high frequency relative to the segmental dynamics, the modulus corresponds to the glassy state,

where the elastic behavior stems from the local mesogen units in the frozen configuration. As high-frequency phonons are the main heat carriers in dielectric materials, the anisotropic phonon propagation will have a significant impact on the directionality of the heat transport. We, therefore, employ the micro-scale temperature wave analysis method to measure the thermal diffusivity parallel and perpendicular to the director of the monodomain LCE.

To address the mechanical and thermal conduction anisotropy and their relationship with the structural ordering, we prepare monodomain main-chain nematic LCEs via thiol-acrylate click chemistry (Fig. 1a)[40]. The highly ordered LCE film offers an ideal model system, where the directional-dependent sound velocities (one longitudinal and two transverse) are resolved by BLS. These measurements enable the complete determination of the elastic moduli and Poisson's ratios (Fig. 1e, f), revealing much higher anisotropy $E_{\parallel}/E_{\perp}$ in LCEs than that in small molecule LCs at GHz frequencies. However, $E_{\parallel}/E_{\perp}$, determined from the macroscopic tensile testing, can be significantly larger than that obtained from BLS, mainly associated with the mesogen orientation at GHz frequencies. The thermal anisotropy exceeds the elastic anisotropy, implying a strong direction-dependent phonon-mean-free path. Given the sensitivity of GHz sound velocities and phonon attenuation to microstructures with a few nematic domains, the proposed semi-soft elasticity and phase transition deformation modes are investigated by strain- and temperature-dependent elasticity, respectively. The former is revealed to preserve the elasticity anisotropy and hence structural ordering as the extension occurs perpendicular to the director, which is in contrast to the observation of a continuous reorientation of directors in most of LCEs at low frequencies[41,42]. For the latter, the continuous decrease in sound velocity with temperature allows us to not only identify $T_{NI}$, but also suggest a different ordering transition feature from the classical first-order one with discontinuity of sound velocity at $T_{NI}$ in LCEs.

## Results

The main-chain LCEs are prepared following our prior work[40] based on an oxygen-mediated thiol-acrylate click reaction (Fig. 1a). The diacrylate mesogen (RM82) and its dithiol-terminated oligomer (RM82-1,3 PDT) undergo the step-growth reactions upon exposure to the UV light, leading to the formation of the polymer network (Methods section). The monodomain LCE film is locally aligned by surface rubbing, while a non-treated polydomain LCE film is prepared as the control. The alignments of both monodomain and polydomain LCE films are confirmed by their cross-polarized optical microscopy images (Supplementary Fig. 1). The monodomain LCE has LC mesogens orientated parallel to the film plane (Fig. 1b, c), where the director **n** points along the principal axis of the polymer shape spheroid. The anisotropy of the polymer backbone, $r = (R_{\parallel}/R_{\perp})^2$, is defined as the ratio of network strand end-to-end distance, $R$, parallel and perpendicular to **n** and equals to the length step ratio ($l_{\parallel}/l_{\perp}$). Albeit no uniquely, the orientation distribution is usually represented by the second Legendre polynomial, i.e., the mean orientation order parameter, $Q$, which is related to the chain anisotropy, $r = (1 + 2Q)/(1 + Q)$, under the assumption of freely joined chains[1].

### Anisotropic elasticity

BLS is a non-contact and non-invasive technique to determine the components of the elasticity tensor under zero strain (see details of the technique in Methods section). According to Christoffel's equation, the elastic constants $C_{ij}$, ($i$ and $j$ running from 1 to 6), determining the different elastic moduli, are coupled with the experimental sound velocities, $c = 2\pi f/q$[39,43]. The sound velocities can be directly obtained from the frequency $f$ of the acoustic phonons in the BLS spectra recorded at different wave vector $\mathbf{q} = \mathbf{k_s}-\mathbf{k_i}$ selected through the orientation angle, $\phi$, relatively to the director **n** of monodomain LCE film (Fig. 1b, c)[44], where $\mathbf{k_i}$ and $\mathbf{k_s}$ are the wave vectors of the incident and scattered lights, respectively, and $q$

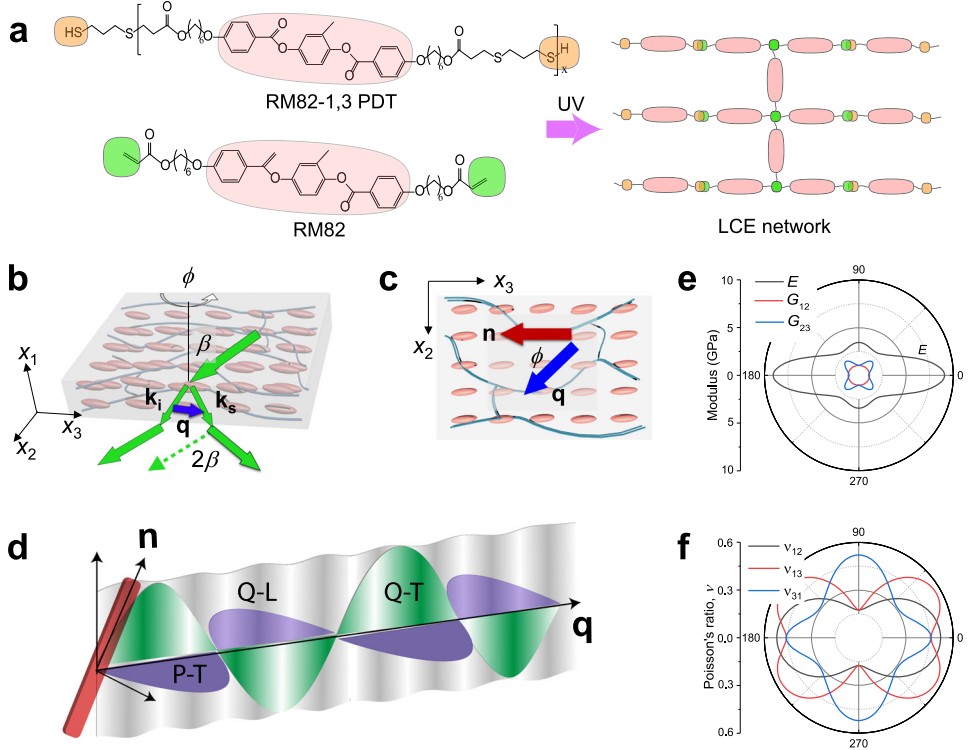

**Fig. 1 | LCE and phonon propagation schematic. a** Scheme of LC oligomer, RM82-1,3 PDT, and diacrylate-based reactive mesogens, RM82, in preparation of LCE network, where the crosslinking sites were introduced via the thiol-acrylate reaction between RM82-1,3 PDT and RM82. **b** Scheme of the transmission scattering geometry used in the BLS experiments. As the scattering angle is twice the incident angle $\beta$, the phonon wave vector $\mathbf{q}=\mathbf{k_i}-\mathbf{k_s}$ is parallel to the $(x_2, x_3)$ film plane where LCs are aligned, $\mathbf{k_i}$ and $\mathbf{k_s}$ being wave vectors of the incident and scattered light, respectively. The $x_1$-axis is normal to the $(x_2, x_3)$ film plane. **c** Rotation of the film around $x_1$ determines the orientation angle $\phi$ between $\mathbf{q}$ and the director $\mathbf{n}$ of the aligned mesogens. **d** Schematic illustration of quasi-longitudinal (Q-L), quasi-transverse (Q-T), and pure-transverse (P-T) acoustic waves with wave vector $\mathbf{q}$ forming an oblique angle with $\mathbf{n}$. Polar diagrams for Young's modulus, $E$, shear moduli, $G_{12}$ and $G_{23}$ in **e** and Poisson's ratio, $v_{12}$ and $v_{31}$ in **f** for the monodomain LCE film.

denotes the magnitude of $\mathbf{q}$. Molecular and polymer liquid crystals are transversely isotropic materials with a symmetry axis (i.e., director $\mathbf{n}$) and an isotropic plane normal to it, thereby reducing the non-zero independent $C_{ij}$ from 21 to 5. Mapping the sound velocities in the plane $(x_2, x_3)$, which contains the symmetry axis, is sufficient to determine the complete elastic moduli: Young's modulus $E_m$ along $m$-direction, the shear modulus $G_{mn}$, shearing in the $m$-$n$ plane, and the Poisson's ratio $v_{mn}$, representing the ratio of contraction in the $n$-direction to the applied extension in the $m$-direction. $m, n$ (=1, 2, 3) denote $x_1$-, $x_2$-, $x_3$-axis of a three-dimensional Cartesian coordinate system (Fig. 1b). In the transmission scattering geometry phonons propagate parallel to the film ($x_2$-$x_3$ plane) with the wavenumber $q = (4\pi/\lambda) \sin\beta$, where $\lambda$(=532 nm) is the wavelength of the laser light in vacuum and $\beta$ is the incident angle (Fig. 1b). At a constant $q$, the orientation angle $\phi$ of the in-plane $\mathbf{q}$ relative to the director $\mathbf{n}$ was tuned from −90° to 90° to prove the symmetric situation around the director $\mathbf{n}$. For monodomain LCE film at a given $\mathbf{q}$, there are a pure-transverse (P-T) acoustic wave for any $\phi$, and quasi-longitudinal (Q-L) and quasi-transverse (Q-T) at $\phi \neq 0$° or 90° as shown in Fig. 1d. We used VH configuration with vertical-polarized (V) incident and horizontal-polarized (H) scattered lights to probe the pure transverse (P-T) acoustic mode in the depolarized VH spectra. The Q-L and Q-T acoustic modes are recorded in the parallel-polarization configuration of either VV or HH spectrum. Access to the sound velocities of Q-L, Q-T, and P-T acoustic modes at various orientation angles $\phi$, allows for the complete determination of the elastic anisotropy as illustrated in Fig. 1e, f for the moduli and Poisson's ratio, respectively. Polarized (VV) and depolarized (VH) BLS spectra for the monodomain LCE were recorded at certain $\phi$'s and $q = 0.0167$ nm$^{-1}$ (incidence angle, $\beta = 45$°) (Fig. 2a and Supplementary Fig. 2). Two distinct

peaks are resolved and represented by Lorentzian lines (green lines), yielding the peak frequency $f$ and the linewidth $\Gamma$ (full-width at half-maximum, FWHM) for each peak. The high- and low-frequency peaks are assigned to Q-L and Q-T modes, respectively. As $\phi$ increases from 0° to 90°, the wave vector $\mathbf{q}$ directs from parallel towards perpendicular to $\mathbf{n}$, and $f_{Q-L}$ decreases from 9.5 to 6.2 GHz in the same direction. The variation of the Q-L frequency, $f_{Q-L}(\phi)$, along with the appearance of Q-T mode suggests strong anisotropy of the sound velocity $c$ and hence elasticity. Due also to the strong birefringence, $\Delta n$, of the monodomain LCE, the Q-L and Q-T modes were respectively observed in the VH and VV BLS spectra as well (Supplementary Fig. 2) and the polarized (VV and HH) BLS spectra at backscattering geometry ($\beta = 0$°) are frequency shifted (Fig. 2b). For the monodomain LCE, $\Delta n = 0.16$ (Supplementary Section 1 and Supplementary Fig. 3) corroborates the notion of a well-ordered LC network, whereas the polydomain LCE is optically isotropic (Supplementary Fig. 4).

Figure 2c illustrates the dependence of sound velocity $c$ on the orientation angle $\phi$ for Q-L, Q-T, and P-T modes (Supplementary Fig. 5). For both Q-L and Q-T, the sound velocities are symmetric about $\phi = 0$°, confirming the transverse isotropy feature of the monodomain LCE film. The angular dependence of $c$ is represented (solid lines in Fig. 2b) by the Christoffel equation (Methods section) using five independent elastic constants ($C_{11}, C_{13}, C_{33}, C_{44}, C_{66}$) as adjustable parameters listed in Supplementary Table 1. Note the absence of dissipation in the Christoffel equation, in contrast to elastic wave propagation at low frequencies comparable to the network Rouse rate[36,45]. Accessing elastic constants $C_{ij}$ allows the estimation of not only the complete elasticities of the monodomain LCE, but also the effective elastic moduli of its polycrystalline structure (Equation 4 in Methods section).

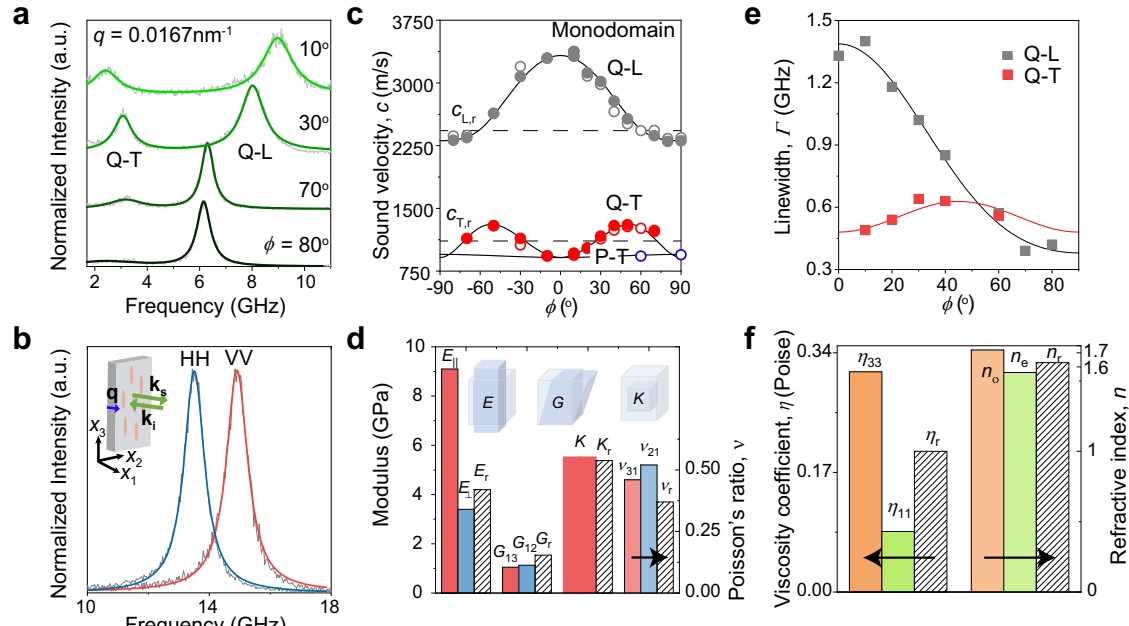

**Fig. 2 | Orientation dependent Brillouin light spectroscopy from monodomain LCE film. a** Polarized BLS spectra of the monodomain LCE film at different orientation angles $\phi$ and a constant wave vector, $q = 0.0167$ nm$^{-1}$, represented by Lorentzian shapes (green lines). The high and low frequency peaks are assigned to the quasi-longitudinal (Q-L) and quasi-transverse (Q-T) phonons, respectively. **b** VV and HH spectra of the monodomain LCE film recorded at backscattering geometry with incident angle $\beta = 0°$. At this geometry, **q** is normal to the $x_2$-$x_3$ plane of the film and hence normal to the director **n**. **c** The sound velocity, $c$, computed from the frequencies of Q-L (gray symbols), Q-T (red symbols) and pure-transverse (P-T, blue symbols) modes, as a function of the orientation angle $\phi$. The solid and open symbols indicate the data obtained from polarized (VV) and depolarized (VH) BLS spectra, respectively. The solid lines denote the representation of sound velocities by Christoffel's equation (Eq. 1 and 2 in Methods section), whereas the dashed lines indicate the experimental values of the longitudinal (LA) and transverse (TA) sound velocities of polydomain LCE film. **d** The elastic properties (Young's modulus $E_\parallel$ and $E_\perp$, shear modulus $G_{12}$ and $G_{13}$, Poisson's ratio $\nu_{31}$ and $\nu_{21}$) for monodomain (red and blue columns) and ($E_r$, $G_r$ and $\nu_r$) for polydomain LCE (shaded columns). The black arrow points to the right axis of the Poisson's ratio, $\nu$. The inset schemes illustrate the physical meanings of Young's, shear and bulk moduli. **e** The linewidth $\Gamma$ of Q-L (gray symbols) and Q-L (red symbols) modes, obtained from polarized BLS spectra in **a** and Supplementary Fig. 2, as a function of the orientation angle $\phi$. The solid lines represent the best fits based on the nonlinear functions in Supplementary Section 2. **f** The viscosity coefficients, $\eta_\parallel = \eta_{33}$ along and $\eta_\perp = \eta_{11}$ normal to the director of polydomain LCE, $\eta_r$ of polydomain LCE, ordinary and extraordinary refractive index ($n_o$ and $n_e$) of monodomain LCE, and refractive index $n_r$ of polydomain LCE. The black arrows point to the left viscosity and to the right refractive index on the left and right axis, respectively.

For the monodomain LCE film, the engineering Young's moduli parallel ($E_\parallel$) and perpendicular ($E_\perp$) to the direction of **n**, shear moduli ($G_{13} = G_{23}$, $G_{12}$), bulk modulus $K$, and Poisson's ratio ($\nu_{31}$ and $\nu_{12}$), deduced from the Equation 3 (Methods section), are shown in Fig. 2d. The anisotropy of Young's modulus, $E_\parallel/E_\perp = 2.63$, is remarkably higher than the compression modulus anisotropy of the small molecule LCs, $M_\parallel/M_\perp \leq 1.15$[20,21,46] also at GHz frequencies[47]. However, the value of $E_\parallel$ is comparable to that of a highly anisotropic molecular (itraconazole) glass with an extreme elasticity anisotropy ($E_\parallel/E_\perp = 2.2$) which decreases with decreasing orientation order parameter $S_m$ due mainly to the decrease of $E_\parallel$[39]. In the direction normal to the director **n**, the value of $E_\perp$ is typical for amorphous glassy polymers. The difference between the sliding $G_{13/23}$ and the torsional $G_{12}$ is within the errors, suggesting similar rigidity in the two main planes. This negligible disparity is in contrast to the large shear anisotropy, $G_{12} \gg G_{13}$, reported for a nacre mimetic hybrid stack due to the large rigidity of the inorganic layer[48]. Both $\nu_{31} = \nu_{32}$ and $\nu_{12} = \nu_{21}$ are typical values of weakly crosslinked elastomer (~0.5)[49], while $\nu_{31}$(=0.46 ± 0.01) is slightly lower compared to the $\nu_{21}$(=0.52 ± 0.03) of the isotropic plane ($x_1$, $x_2$); stretching along the $x_1$-axis leads to smaller contraction along $x_3$-axis than along $x_2$-axis[50]. Interestingly, the LCE may be compressible ($\nu_{13(23)} + \nu_{12(21)} < 1$) at a normal strain to director **n**, violating the volume conservation assumption that is normally used for estimation of Poisson's ratio of LCEs or any rubber materials[38,51]. The Poisson's ratio of anisotropic LCE should have no boundaries[52] unlike the isotropic elastomers ($0 < \nu \leq 0.5$). In fact, compressibility was reported in main-chain chiral nematic LCE, where both $\nu_{13}$ and $\nu_{12}$ were found larger than 0.5

allowing for a significant volume change by applying a small transverse deformation[15].

The Young's modulus obtained from BLS at GHz frequencies falls in the GPa range, which is three orders of magnitude higher than the corresponding $E$ from the tensile testing[33], that is, $E_{\parallel(\perp),\ \text{tensile}} = 10(3)$ MPa estimated from the stress-strain curve in the initial linear regime for the monodomain LCE[14]. Even much lower $G_{\parallel(\perp),\text{tensile}} \approx 90(40)$ kPa was reported for a monodomain LCE, when the director rotation under stress was prohibited[32]. So the anisotropic elasticity at low frequencies in the rubbery state assumes low moduli values[32,53,54] In fact, it has been reported from the uniaxial tensile test at different strain rates (0.1–100 Hz) and temperatures (−20 to 70 °C) in a similar main-chain LCE, that $E$ increases from ~1 MPa at the high temperatures (frequencies in Hz) to ~3 GPa at low temperatures (frequencies in 10 kHz) obtained through t-T superposition. This is due to slow network relaxation frequencies (about 0.5 MHz at ambient temperature)[34] and not due to the much faster local mesogen dynamics related to the glass transition temperature (~278 K). This finding motivates the elucidation of the mesogenic orientation in the network to the mechanical properties along the director in an isothermal experiment but at frequencies that are sufficiently high to exclude the network contribution. However, there was no report on the elastic moduli and their anisotropy at GHz, which is necessary to describe the mesogenic orientation-dependent mechanical response in LCEs. The inherently high-frequency BLS experiments presented in this study aim to fill this gap. Similarly, the shear moduli from BLS fall in the GPa range, typical for glassy (elastic) polymers[55]. As a reference, the shear modulus of a typical isotropic

viscoelastic polymer network such as PDMS is, $G = E/[2(1 + \nu)] \approx 1.4$ MPa, at low frequencies, corresponding to the elastomeric state[49,55]. Recently, through three tensile testing experiments in directions that are parallel, perpendicular, and oblique to the director field of a LCE, $G_{23} \approx G_{12} \approx 2.2$ MPa were reported[38]. Since the strain rate ($-0.01\,\text{s}^{-1}$) in the tensile test experiments is very low, the moduli correspond to their static value, i.e., at the limit of low frequencies[34]. In contrast, BLS carried out at high frequencies reports the elastic moduli of the "frozen" LCEs, thus, assuming much higher values than the corresponding viscoelastic moduli[34]. In fact, the polymer network strand dynamics controlling the LCE viscoelasticity fall in the intermediate MHz frequency range[56], and the large discrepancy in the values of $E$ clearly reflects its strong frequency dependence for the viscoelastic LCEs[34]. In addition, the low-frequency moduli are proportional to the crosslink density which has a weak effect on the high-frequency counterparts. In this context, the recently reported storage $E'$ from t-T superposition of dynamic frequency scan viscoelastic tests displays a crosslinking dependence (from 1 to about 3 GPa over the real frequency range of 0.01–200 Hz)[35]. While this isotropic value is comparable to $E_\perp$ in Fig. 2d, the large error of the t-T superposition and more importantly the assumption of a robust structure over the broad scanned T-range renders the reported strong crosslinking dependence questionable[35]. Notably, the t-T superposition applies for either nematic or isotropic states but does not hold across the phase transitions[34–37].

Besides the vastly different high (BLS) and low (tensile test) frequency modulus values, the anisotropy, $E_{\parallel,\text{tensile}}/E_{\perp,\text{tensile}} = 3.3$[14,22] is higher than the elastic anisotropy value ($E_\parallel/E_\perp = 2.63$) measured from the present BLS study (Fig. 2d). For a similar LCE, the low-frequency $E_{\parallel,\text{tensile}}$ and $E_{\perp,\text{tensile}}$ sensitively depend on the surface rubbing treatment to induce the alignment; that is $E_{\parallel,\text{tensile}}$ increase from 10.3 to 16 MPa and $E_{\perp,\text{tensile}}$ decreases from 4.2 to 1.2 MPa, respectively, with anisotropies $E_{\parallel,\text{tensile}}/E_{\perp,\text{tensile}}$ ranging from 2.45 to 13.3[54,57]. In the same context, the addition of a minute amount of Au nanorods (less than 0.02 vol%) boosts both $E_{\parallel,\text{tensile}}$ and $E_{\perp,\text{tensile}}$ by almost 100% with a moderate effect on the mechanical anisotropy ($E_{\parallel,\text{tensile}}/E_{\perp,\text{tensile}}$)[54]. Compared to the mesogen size ($-2-3$ nm), the relatively long ($-18$ nm) nanorods can increase the effective crosslink density of the aligned network strands but has virtually no impact on the high-frequency $E_\parallel$ and $E_\perp$ (Supplementary Fig. 6). In fact, the elastic constant anisotropy ($C_{33}/C_{11} \approx 2.2$) of polymer liquid crystals[58] (no crosslinks) at hypersonic frequencies is very similar to $C_{33}/C_{11} = 2.1$ of the LCE. We are, therefore, left to assert that the elastic anisotropy $E_\parallel/E_\perp$ is the reliable quantity related to the nematic mesogen orientation in LCEs. An estimate of the chain anisotropy $r$ is usually reported from tensile testing perpendicularly to the stretching direction using the two critical extension ratios, $\lambda_1 < \lambda_2$, at the crossover initially from isotropic rubber to soft fluid, in which the director rotation takes place, and then crossover to strain hardening regime (Supplementary Fig. 7a)[59]. As already mentioned, the microscopic order parameter, $Q = (r-1)/(r+2)$, can be computed from $r$ under the assumption of a freely joined chain[1]. The reported $r$ of the monodomain LCE varies between 2.6 ($Q = 0.35$) and 5.5 ($Q = 0.6$)[4,7,50,59]. An additional estimate of $r$ ($=\lambda_T^3$) can be directly obtained from the thermally induced contraction, $\lambda_T$, of LCE along its director in the isotropic state. For the present case, $r = 7.1 \pm 0.17$[54] and hence $Q = 0.67 \pm 0.08$. For a second optimally aligned LCE film (named LCE-II) prepared by extensive rubbing, $E_{\parallel,\text{tensile}}/E_{\perp,\text{tensile}} \approx 12.3$, and the tensile testing normal to the director $\mathbf{n}$ (Supplementary Fig. 7) yields high $\lambda_2$ value and $r = (\lambda_2/\lambda_1)^2 = 9.7 \pm 1.5$ implying very high orientation with $Q = 0.74 \pm 0.03$. For a similar LCE film[57] with $E_{\parallel,\text{tensile}}/E_{\perp,\text{tensile}} \approx 13.3$, the length step ratio $r$ amounts to $r = 5.8 \pm 0.2$ and $Q = 0.62 \pm 0.02$. While a unique relation between $r$ and $E_{\parallel,\text{tensile}}/E_{\perp,\text{tensile}}$ should exist, the available data suggest that the relation between deformation and $r$ should be questioned[60] and the coupling between chain conformation anisotropy ($r$) and nematic order-parameter ($Q$) is strongly system dependent[61]. Remarkably, the

present and LCE-II samples display very similar high-frequency elasticity anisotropy $E_\parallel/E_\perp$ (=2.8) and engineering moduli (Supplementary Fig. 7 and Supplementary Table 3), in contrast to the very different $E_{\parallel,\text{tensile}}/E_{\perp,\text{tensile}}$ at low frequency.

The phonon attenuation is also of fundamental importance for LCE properties in view of its anisotropy. There are several sources of elastic wave dissipation in solids such as thermal conduction and interaction with thermal phonons, yielding complex elastic constants $C'_{ij} = C_{ij} + i\omega\eta_{ij}$, where $\eta_{ij}$ are the components of the viscosity tensor and $\omega = 2\pi f$ is the angular frequency. The viscosity constants, $\eta_{ij} = \frac{2\pi\rho\Gamma}{q^2}$, are associated with the full-width at half-maximum $\Gamma(\phi)$ of the Brillouin peak, where $\rho$ is the material's density. According to the hydrodynamic theory for LC[62], the viscous effect is also anisotropic and $\eta_{ij}$ has five independent constants. The eigenvalue of the viscosity tensor could be determined (Supplementary Section 2) in the same manner as $C_{ij}$. Figure 2e displays the experimental $\Gamma(\phi)$ of Q-L and Q-T modes obtained from the BLS spectra (Fig. 2a and Supplementary Fig. 2) along with the theoretical representation denoted by the solid lines; the nonlinear Chi-square fitting includes four adjustable $\eta_{ij}$ (Supplementary Table 2). Here, the extremes of $\Gamma_{\text{Q-L}}$ at $\phi = 0^\circ$ and $90^\circ$ feature an anisotropy, $\eta_{33}/\eta_{11} = 3.7$ (Fig. 2f), which is noticeably stronger than the elasticity anisotropy, $C_{33}/C_{11} = 2.1$, in the same sample. At GHz regime, the phonon attenuation is associated with the local viscosity-induced dissipation process[23,63] related to local density dynamics. As compared to molecular LCs with $\eta_{33}/\eta_{11} = -2$[21,64], viscosity anisotropy is apparently larger in the present LCE. For both molecular LC and LCE, however, the viscosity anisotropy exceeds the elasticity anisotropy, suggesting a higher sensitivity of $\eta_{33}/\eta_{11}$ on the mean orientational order than $C_{33}/C_{11}$.

## Elasticity response to external stimuli

The facile reorientation of LC mesogens and the strain-induced macroscopic shape changes offer LCEs tremendous potentials as reversible shape memory materials. The best-known features are the semi-soft elasticity (SSE) and the LC phase transition both associated with the transformable orientation order under mechanical and thermal stimuli, respectively[59]. SSE is manifested as a plateau-like region after the initial linear elastic response in the stress-strain curve when the uniaxial strain is applied normal to $\mathbf{n}$[1,41,50,59]. This low stress-cost in-plane deformation is associated with the continuous reorientation of $\mathbf{n}$ in the plane with low (soft) shear resistance ($G_{13}$) resembling liquids. The rotation of the network strand shape proceeds with no space distortion and energy cost and ideally LCEs enable absorption of strain energy at constant stress[22]. Upon completion of the director rotation, from initially perpendicular to finally parallel to the director $\mathbf{n}$, the stress increases with strain again. An alternative response is a sharp director rotation at a critical strain in analogy to the mechanical Fréedericksz transition (MFT) reported for a few acrylate-based LCEs[1,56,59,60,65]. We are not aware of a tensile load curve displaying an MFT, which is so far revealed through tracking of the director rotation in LCEs by optomechanical and scattering techniques throughout tensile testing[32,59,60].

All published reports on the mode of mechanical deformation of LCEs are based on low-frequency loading with director $\mathbf{n}$ orientation. To investigate the corresponding high-frequency response and local nematic order, we trace the sound velocities $c_\parallel$, $c_\perp$ for two applied strains, parallel ($\mathbf{u}\parallel\mathbf{n}$) and normal ($\mathbf{u}\perp\mathbf{n}$) to the director as illustrated in Fig. 3a. At $q = 0.0167\,\text{nm}^{-1}$, the evolution of polarized (VV) BLS spectra with two orthogonal strains is depicted in Fig. 3b. A single peak in the BLS spectrum, assigned to the Q-L mode, is expected for both cases ($\phi = 0^\circ$ or $90^\circ$) and the corresponding sound velocities ($c_\parallel$, $c_\perp$) and linewidths ($\Gamma_\parallel$, $\Gamma_\perp$) are displayed in Fig. 3c, d, respectively. Both physical parameters appear nearly constant under the uniaxial deformation, implying robust elastic anisotropy (and hence orientation order) to the stretching of the aligned LCE at ambient conditions up to the fracture strain, $s_F$ $-0.5$.

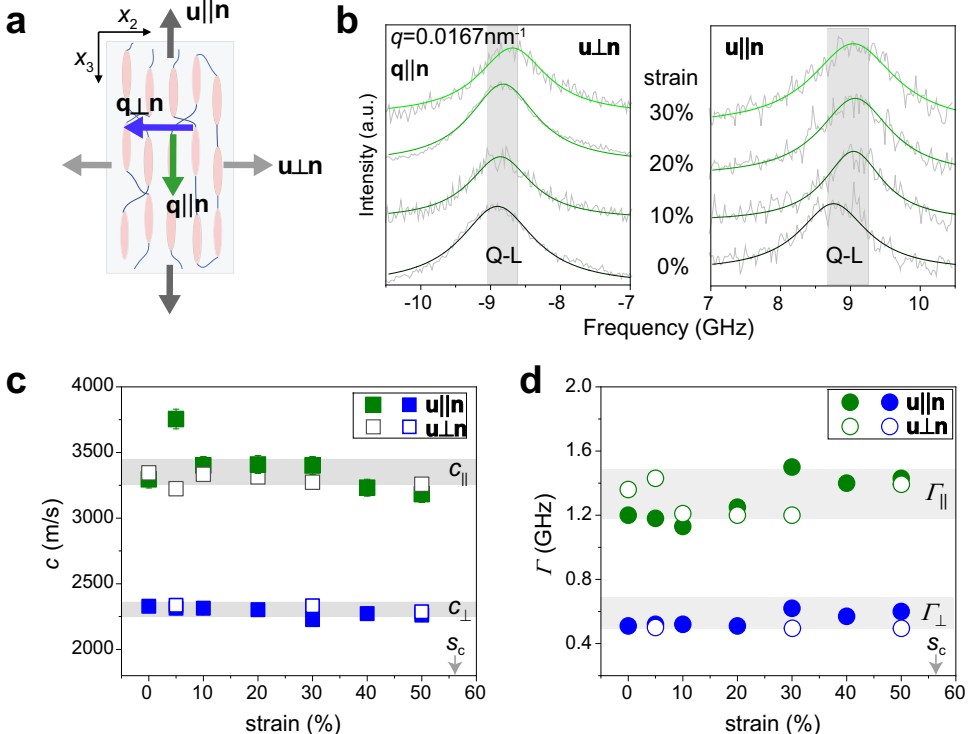

**Fig. 3 | Strain-dependent elasticity. a** Scheme of the aligned LCE under uniaxial stretching. The uniaxial stretching direction (gray arrows) is either parallel (**u**∥**n**) or perpendicular (**u**⊥**n**) to the director **n** in the $x_2$-$x_3$ plane, and the phonons at orthogonal directions (**q**∥**n**, **q**⊥**n**) are selectively probed. **b** Evolution of polarized BLS spectra for **u**⊥**n** (left) and **u**∥**n** (right), respectively, at a given **q** being parallel to **n**. The BLS spectra (gray lines) are well represented by Lorentzian shape (green lines) assigned to Q-L mode. **c** and **d** show the sound velocity $c$ (squares) and linewidth $\Gamma$ (circles) of Q-L modes as a function of strain, respectively. The green and blue symbols denote phonon propagation parallel (∥) and normal (⊥) to the director, **n**, respectively. The solid and open symbols represent the stretching for **u**∥**n** and **u**⊥**n**, respectively. The two gray arrows indicate the positions of critical strain $s_c \approx 0.56$.

While for **u**∥**n** stretching the robust $c$ with strain is an anticipated finding, for the perpendicular (**u**⊥**n**) deformation, the soft elasticity mode, the experimental observation of a virtually constant anisotropy ($c_\parallel/c_\perp$) in Fig. 3c appears counterintuitive as the director field would have been continuously reoriented toward the direction of the stress and hence manifested in a decrease of $c_\parallel/c_\perp$. The corresponding tensile load curve of the latter case presented the plateau-like region with little stress increase when $s > 0.25$, which is evidence of SSE-like deformation[54]. On the contrary, the robust hypersonic sound velocity anisotropy $c_\parallel/c_\perp$ upon perpendicular (**u**⊥**n**) stretching in Fig. 3c conforms to an MFT-like response with constant director orientation normal to the direction of stress; for an SSE deformation of our LCE sample, director rotation should have already started above $s > 0.25$. Instead, MFT-like deformation defines a stable ordering in this plateau-like region in which reorientation occurs sharply at a critical strain $s_c > 0.25$ (gray arrow in Fig. 3c, d)[59]. For our LCE, the stress–strain curve shows it breaks in the plateau-like region and hence $s_f < s_c$ rationalizing a virtually constant $Q$ with the director still normal to the stress axis. Our LCE has $r \approx 3.8$, the director rotation should be expected to occur at $s_c = r^{1/3} - 1 \approx 0.56$[1]. Albeit our LCE breaks at $s_f$ just before the estimated critical strain $s_c$ for an abrupt director rotation towards the stretching direction, the MFT-like response is further supported by the Q-L phonon linewidths plotted vs strain in Fig. 3d. The phonon attenuation anisotropy ($\Gamma_\parallel/\Gamma_\perp$) remains, like the sound velocity anisotropy, within the experimental error constant throughout the stretching process. Since the linewidth anisotropy, $\Gamma_\parallel/\Gamma_\perp = \eta_{33}/\eta_{11}$, the phonon attenuation anisotropy also follows the local mesogenic orientation and hence Fig. 3d implies a robust director rotation normal to the strain axis (**u**⊥**n**) which again is compatible for an MFT-like response. This notion is also corroborated with invariable sound velocities and linewidth of P-T mode under both parallel (**u**∥**n**) and

normal (**u**⊥**n**) stretching in Supplementary Fig. 8. So far, the reported LCEs featuring MFT are all acrylate-based elastomers[51,59,60]. Whether the chemical structure is critical to the MFT behavior remains unclear and requires more experimental and theoretical investigations.

The nematic-to-isotropic transition causes a pronounced elasticity softening and decrease of elasticity anisotropy as a function of temperature. We recorded the polarized BLS spectra with **q** parallel to director **n** at different temperatures shown in Fig. 4a and Supplementary Fig. 9 for monodomain and polydomain LCE films. Figure 4b displays $c_\parallel$ of monodomain, and $c_r$ of polydomain LCE films, which decrease almost linearly with temperature and $c_\parallel$ merges $c_r$ at $T_{NI} = 396$ K. The latter value is consistent with $T_{NI} = 390$ K measured from the differential scanning calorimetry (Supplementary Fig. 10). The opacity of both LCE films above $T_{NI}$, which may result from the multiple scattering from the discernible microscopic domains[40], precludes the recording of $q$-dependent BLS spectra. The ratio, $c_\parallel^2/c_r^2$, estimated from the linear representations of both $c_\parallel(T)$ and $c_r(T)$, monotonically decreases from 1.67 to 1 with temperature as shown in the inset of Fig. 4b. It should be noted that the transition of $c_\parallel$ to $c_r$ at $T$-$T_{NI}$ is rather broad due to crosslinking constraints, which is different from the discontinuous first-order transition reported for some molecular liquid crystals[21,66]. Besides the sound velocity, the acoustic attenuation coefficient $\mu(=\pi\Gamma/c)$ can identify the phase transition. The linewidth $\Gamma$ obtained from the Lorentzian representation of the BLS spectra increases with the temperature being clearly stronger for the monodomain ($\Gamma_\parallel$) than the polydomain ($\Gamma_r$) LCE as depicted in Supplementary Fig. 11. This temperature-dependence is much stronger for the acoustic attenuation coefficient ($\mu=\pi\Gamma/c$) with the increase of $\mu_\parallel$ being much steeper than for $\mu_r$ as shown in Supplementary Fig. 12. This behavior is not macromolecular property as it was observed for a molecular LC by ultrasonic experiments[64].

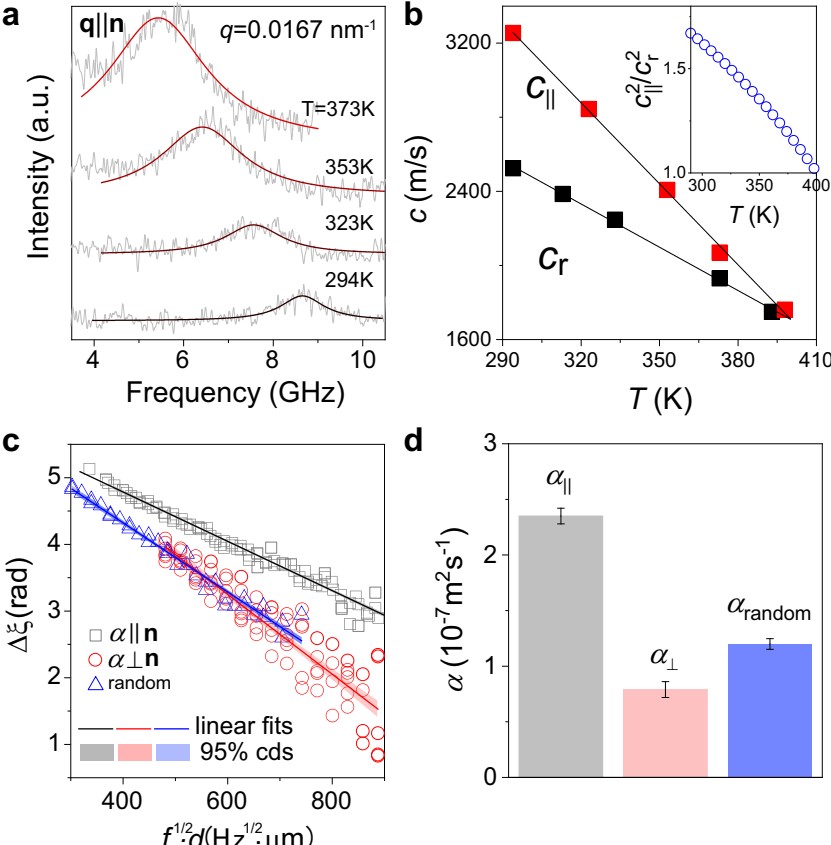

**Fig. 4 | Temperature-dependent elasticity and anisotropic thermal diffusivity.**
**a** Evolution of polarized BLS spectra with the temperature at a given **q** parallel to **n**.
**b** The temperature-dependence of $c_\parallel$ in monodomain LCE and $c_r$ in polydomain LCE along with their ratio shown in the inset. The solid lines denote linear representation of the temperature-dependent sound velocities. **c** The phase delay $\Delta\xi$ of the heat waves traversing the LCE films with a thickness $d$ as a function of $f^{1/2}\cdot d$, where $f$ is the frequency of the heat source. The gray and red symbols denote the thermal diffusivity $\alpha$ along and normal to the director **n** of monodomain LCE film, respectively, whereas blue symbols denote $\alpha_{random}$ in the polydomain LCE film. The $\Delta\xi$ drops linearly with $f^{1/2}\cdot d$ (solid lines, shaded area: 95% confident bands). The thermal diffusivity $\alpha$ computed from the slopes of the linear representation of $\Delta\xi$ vs $f^{1/2}\cdot d$ is shown in **d**.

## Anisotropic heat conduction

Phonons are responsible for the heat diffusion in dielectric matter and their anisotropic propagation can also impact the directionality of the heat transport. In principle, the nanoscaled structures impact both the high-frequency sound velocities and heat conductivity, and hence the corresponding anisotropies in heat conductivity can be an additional measure of their sensitivity to the orientation order parameter. For a molecular LC, the mechanical anisotropy was found to be smaller than the thermal anisotropy in the LC phase[21]. In this study, the relation of elasticity anisotropy to the anisotropic heat conduction was investigated employing the micro-scale temperature wave analysis method to measure the thermal diffusivity, $\alpha_\parallel$ and $\alpha_\perp$, respectively, parallel and perpendicular to the director **n** of monodomain LCE as described in Methods Section. When the generated temperature wave with a frequency $f$ crosses the distance $d$, its phase delay $\Delta\xi$ is correlated to the thermal diffusivity $\alpha$ by $\Delta\xi \sim (\pi f/\alpha)^{1/2}d$[67]. Fig. 4c shows $\Delta\xi$ as a function of ($f^{1/2}d$) as heat propagates along and normal to the director **n** in the monodomain and polydomain LCE films, respectively. The corresponding thermal diffusivities, $\alpha_\parallel = (2.35 \pm 0.17)~10^{-7}$ m$^2$/s, $\alpha_\perp = (0.79 \pm 0.07)~10^{-7}$ m$^2$/s and $\alpha_{random} = (1.20 \pm 0.03)~10^{-7}$m$^2$/s, are shown in Fig. 4d. Using the experimentally measured LCE density, 1.25 g/cm$^3$, and specific heat, 1.57 J g$^{-1}$ K$^{-1}$, we obtain the heat conductivities $\kappa_\parallel = 0.46$ W m$^{-1}$ K$^{-1}$ and $\kappa_\perp = 0.16$ W m$^{-1}$ K$^{-1}$, in the typical range for isotropic polymers. The thermal anisotropy, $\kappa_\parallel/\kappa_\perp = 3.0$ is slightly higher than the corresponding elasticity anisotropy ($E_\parallel/E_\perp = 2.63$), but similar to the observation from a molecular LC[21].

In the kinetic theory, the thermal diffusivity is related to the phonon group velocity and phonon-mean-free paths ($\Lambda$) and the use of effective properties leads to $\alpha = \Lambda(c_L + 2c_T)/3$. In the case of the monodomain LCE, $\Lambda$ is found to be direction-dependent assuming an anisotropy, $\Lambda_\parallel/\Lambda_\perp = 2.4$. This ratio is smaller than the geometric aspect ratio of rigid core (length/width = 1.58 nm/0.424 nm = 3.8), whereas in the case of a molecular LC (5-n-octyl-2-(4-n-octyloxy-phenyl)-pyrimidine), $\Lambda_\parallel/\Lambda_\perp$ is comparable with the aspect ratio ($\approx 2.9$) of the rigid core[21]. In the case of LCE, the rigid core is part of the network strands and the anisotropy of the rigid LC core no longer represents the $\Lambda_\parallel/\Lambda_\perp$ anisotropy, which suggests a lower aspect ratio of an effective rigid core[68,69]. For both molecular LC and LCE, the thermal transport is anisotropic, but the network structure of the latter can be an additional parameter that controls $\Lambda_\parallel/\Lambda_\perp$ and hence the heat transport anisotropy.

Based on the limited literature[28-30] on anisotropic heat transport in LCEs, the heat conductivity anisotropy $\kappa_\parallel/\kappa_\perp$ varies over a large range and the relation to local mesogenic order parameter $S_m$ is rather ambiguous. For a highly crosslinked main-chain liquid crystalline network[29] of similar acrylate chemistry, $\kappa_\parallel = 0.34$ W m$^{-1}$ K$^{-1}$ and $\kappa_\parallel/\kappa_\perp \approx 1.5$ while $S_m = 0.6$. For a main-chain LCE that is homeotropically anchored on the surface with $S_m = 0.73$, the out-of-plane heat conductivity, $\kappa_\parallel = 3.56$ W m$^{-1}$ K$^{-1}$ is extremely high, leading to a heat transport anisotropy $\kappa_\parallel/\kappa_\perp \approx 15$[28]. LCEs of slightly different chemistry possess $\kappa_\parallel \approx 2.3$ W m$^{-1}$ K$^{-1}$ with $\kappa_\parallel/\kappa_\perp$ between 6 ($S_m = 0.61$) and 11 ($S_m = 0.72$)[30]. A plausible explanation of the apparent ill-defined relation between heat conductivity anisotropy and $S_m$ is the difference between the intrinsic $\kappa_\parallel$ and $\kappa_\perp$ for $S_m = 1$[39]. A structure-based

description of thermal anisotropy requires not only the local meso-genic order parameter $S_m$, but also the non-mesogenic polymer strands in the network, which can impact the phonon-mean-free paths ($\Lambda$) and the thermal transport properties.

## Discussion

The elasticity is distinctly different at high frequencies probed by BLS from the tensile testing at low frequencies. The three orders of mag-nitude difference between the high and the low-frequency Young's moduli is anticipated due to the elastomeric state of the LCEs with slow (about 0.5 MHz at ambient temperature) network relaxation fre-quencies. However, the finding of a higher low-frequency (tensile testing) elastic anisotropy, $E_{\parallel,tensile}/E_{\perp,tensile}$, than the high-frequency (BLS) elastic anisotropy, $E_{\parallel}/E_{\perp}$, is unanticipated. It suggests that the contribution of the mesogenic orientation at GHz is different from tensile testings conducted at low frequencies, where LCE deforms in response to the orientational change of the network strands. In fact, the two LCE samples of the present study based on the same cross-linking density but different surface alignment treatments, exhibit very different, by a factor of five, $E_{\parallel,tensile}/E_{\perp,tensile}$ (between 2.45 and 13) due to different shape anisotropy of the polymer spheroid expressed in the value of the chain anisotropy parameter, $2.6 < r < 9.7$ (Supplementary Fig. 7)[14,54,57]. At high frequencies, however, both LCE films display very similar elasticity anisotropy $E_{\parallel}/E_{\perp}$ ($\approx 2.6$) and engineering moduli (Sup-plementary Fig. 7 and Supplementary Table 3). Since the elasticity anisotropy $E_{\parallel}/E_{\perp}$ should also depend on the local order parameter $S_m$, as revealed in the case of orientated nematic glasses[39], the latter does not necessarily follow $r$ or $Q$ in the case of LCEs. On the premise of a single orientation distribution function[70], $S_m$ should be lower than $Q$, implying on average the segmental mesogenic orientation has higher orientation angles. This is an unexpected finding, uniquely attributed to the LCE high-frequency elasticity.

The strain-induced macroscopic shape changes of LCEs under deformation perpendicular to the order parameter correspond to the best-known SSE associated with a transformable orientation order[50]. Consequently, in a tensile load, the director rotation occurs continuously from initially perpendicular to finally parallel to the direction of strain. Conversely, a sharp director rotation at a critical strain, referring to the MFT, is reported only when monitoring the director rotation by optomechanical and scattering techniques[59,60]. The published literature on the mode of mechanical deformation of LCEs is based on low-frequency loading with **n** orientation. For the corresponding high-frequency response, the trace of the sound velocities $c_{\parallel}$, $c_{\perp}$ (Fig. 3c) for strain normal ($\mathbf{u} \perp \mathbf{n}$) to the director (Fig. 3a) corroborates the notion of an MFT deformation mode. The seemingly contradicting results probing the director rotation from the shape of load curve[54] (Supplementary Fig. 7) and from the sound velocity anisotropy (Fig. 3c) and phonon attenuation (Fig. 3d) left us to conclude that the former is not sufficient to determine the deformation mode. A similar observation was reported for a tensile testing and optomechanical probing of the director rotation[59].

As anticipated from the phonon propagation anisotropy, the pre-sent LCE exhibits anisotropic heat transport characterized by the heat conductivities, $\kappa_{\parallel} = 0.46$ W m$^{-1}$ K$^{-1}$ and $\kappa_{\perp} = 0.16$ W m$^{-1}$ K$^{-1}$ and aniso-tropy, $\kappa_{\parallel}/\kappa_{\perp} \approx 3$ being slightly higher than the corresponding elasticity anisotropy ($E_{\parallel}/E_{\perp} = 2.6$). Based on the limited literature[28–30], the relation of the anisotropy $\kappa_{\parallel}/\kappa_{\perp}$ varying over a large range and the mesogenic order parameter is rather ambiguous. In a structure-based description of the thermal anisotropy, we should take into account the direction-dependent phonon-mean-free paths. In analogy to the photo-mechanical response of LCEs[71,72] that leads to bending, inhomogeneous deformations can benefit from non-uniform heating in the absence of dyes harnessing the anisotropic thermal conduction in LCEs.

In summary, our results pave the way to unravel the complexity of the frequency-dependent mechanical responses in LCEs. We show that

superior nanoscale mechanical properties can be achieved by max-imizing the local mesogenic orientation, which should also benefit the fast heat conduction along the director. The different length scales can be at the origin of the distinct response to uniaxial load-deformation normal to the director at low and high frequencies. The fundamental understanding of thermoelasticity anisotropy presented here will offer new sights to tailor the macroscopic responses via the interplay of chemical structures, molecular orientation, and network structures that are applicable in broad fields, especially when operated at high frequencies for applications such as the 5G broadband wireless com-munications. At such high frequencies, how to transport sound and heat efficiently will be critical to improve the device's reliability.

## Methods

### Materials

LC mesogen (1,4-bis-[4-(6-acryloyloxyhexyloxy) benzoyloxy]-2-methyl-benzene (RM82) was purchased from Wilshire Technologies. Chain extender 1,3-propane dithiol (1,3-PDT), catalyst 1,8-diazabicyclo(5.4.0) undec-7-ene, inhibitor butylated hydroxytoluene photoninitiator 2,2-dimethoxy-2-phenylacetophenone (DMPA), and poly(vinyl alcohol) (Mowiol 20-98, $M_w$ -125,000) were purchased from Sigma-Aldrich. Dichloromethane and hydrochloric acid were purchased from Fisher Scientific. Polydimethylsiloxane (PDMS) Sylgard 184 Elastomer Kit was purchased from Dow Corning Corporation. All chemicals were used as received without further purification.

### Fabrication of liquid crystal elastomer films

The fabrication process includes two steps, which are the synthesis of thiol-terminated LC oligomers and the thiol-acrylate reaction between LC oligomers and diacrylate based reactive mesogens to introduce cross-linking sites in the networks[40]. In the first step, the commercially available diacrylate, 1,4-bis-[4-(6-acryloyloxyhexyloxy) benzoyloxy]-2-methylbenzene (RM82), was reacted with a short-chain dithiol, 1,3-propanedithiol (1,3 PDT), through a base-catalyzed click reaction to obtain the thiol-terminated oligomers, referred as RM82–1,3 PDT. In the second step, RM82 and RM82-1,3 PDT were mixed at 1:1 molar ratio with 2 wt% photoinitiator, DMPA. The mix-ture was heated and stirred at 120 °C for 5 min, and then infiltrated via capillary force into a 50 μm thick glass cell at 120 °C, which had an epoxy layer on both sides with rubbing treatment from a piece of velvet cloth. The glass cell with the infiltrated mixture was then cooled to 50 °C and annealed for 1 h before UV exposure at room temperature with a total dosage of 20 J/cm$^2$. The crosslinked LCE film was removed from the glass cell to obtain the final free-standing film. As a control sample, the polydomain LCE films were prepared using the same glass cell but without any surface rubbing treatment.

### Brillouin light spectroscopy (BLS)

BLS probes the light scattered from the thermally active phonon with wavevector $\mathbf{q} = \mathbf{k_i} - \mathbf{k_s}$ and frequency $f = cq/2\pi$, where $\mathbf{k_i}$ and $\mathbf{k_s}$ are wavevectors of incident and scattered lights, and $c$ is sound velocity. The BLS measurements were performed using a six-pass tandem Fabry–Perot interferometer in conjunction with the Nd/YAG laser ($\lambda = 532$ nm) mounted on a goniometer, allowing for $q$-dependent measurements. The typical BLS spectrum includes a doublet with frequency shift $f$ relatively to the central ($f = 0$) Rayleigh line. The experiments were performed in the transmission, reflection, and backscattering geometries. In the transmission geometry, the $q(= 4\pi\sin\beta/\lambda)$ is directed in the film plane whereas $q(= 4\pi\sqrt{n^2 - \sin^2\beta}/\lambda)$ is normal to the film plane in the reflection and backscattering geometries, where the incident angle $\beta$ is 180° in the backscattering geometry. For anisotropic systems, the polarizations of longitudinal and transverse acoustic waves are not pure, but hybrid quasi-waves with displacement field direction being neither

parallel nor perpendicular to the propagation direction. The quasi-longitudinal (Q-L) and quasi-transverse (Q-T) phonons are detectable in the VV spectra, while pure-transverse (P-T) is active in the VH spectra. The VV(VH) denotes the combined polarization of the incident and scattering lights, selected by the input polarizer (V) and output analyzer (V or H); V(H) denotes vertically(horizontally) polarized light with respect to the scattering plane defined by the $\mathbf{k_i}$ and $\mathbf{k_s}$. For temperature-scan experiments, the temperature increases at a slow speed of 0.1 °C/min and BLS spectra were collected after an isothermal equilibration of the whole setup for 10–20 min. The uncertainty of the sound velocities is within 2%.

For the determination of anisotropic elasticities of monodomain LCE film, we employed transmission geometry to examine in-plane phonon propagation at a constant $q$ in Fig. 1b. The orientation angle $\phi$ of the in-plane $\mathbf{q}$ relative to the director $\mathbf{n}$ (Fig. 1c) was tuned in the range of −90° to 90° to prove the symmetric situation around the director $\mathbf{n}$. At a given $\mathbf{q}$ and $\phi$, (quasi-) longitudinal and (quasi- and pure-) transverse acoustic waves are measured separately by employing different polarization combinations of the incident and scattered lights (Figs. 2b and S2). The sound velocities, $c(=2\pi f/q)$, as a function of orientation angle $\phi$ was resolved, which can hence determine the elastic stiffness tensor according to Equation 2. The monodomain LCE film is transversely isotropic with the symmetry $x_3$-axis, and the corresponding elastic stiffness tensor $\mathbf{C}$ (in the Voigt notation) includes five independent elastic constants, $C_{11}$, $C_{13}$, $C_{33}$, $C_{44}$, and $C_{66}$ (Eq. (1)).

$$\mathbf{C} = \begin{bmatrix} C_{11} & C_{11}-2C_{66} & C_{13} & & & \\ C_{11}-2C_{66} & C_{11} & C_{13} & & & \\ C_{13} & C_{13} & C_{33} & & & \\ & & & C_{44} & & \\ & & & & C_{44} & \\ & & & & & C_{66} \end{bmatrix} \quad (1)$$

The expressions of sound velocities of quasi-longitudinal (Q-L), quasi-transverse (Q-T), and pure-transverse (P-T) phonons and elastic constants are shown in Eq. (2):

$$c_{Q-L} = \sqrt{\frac{-A_1 + \sqrt{A_1^2 - 4A_2}}{2\rho}} \quad (2a)$$

$$c_{Q-T} = \sqrt{\frac{-A_1 - \sqrt{A_1^2 - 4A_2}}{2\rho}} \quad (2b)$$

$$c_{P-T} = \sqrt{\frac{A_3}{2\rho}} \quad (2c)$$

where $\phi$ is the orientation angle between wavevector $\mathbf{q}$ and director $\mathbf{n}$, $\rho = 1250$ kg/m³ is the density of LCE films, and

$$A_1 = -\left(\sin^2\phi\, C_{11} + \cos^2\phi\, C_{33} + C_{44}\right) \quad (2d)$$

$$A_2 = \sin^4\phi\, C_{11}C_{44} + \sin^2\phi\cos^2\phi\left(C_{11}C_{33} - C_{13}^2 - 2C_{13}C_{44}\right) + \cos^4\phi\, C_{33}C_{44} \quad (2e)$$

$$A_3 = \sin^2\phi\, C_{66} + \cos^2\phi\, C_{44} \quad (2f)$$

The engineering elastic properties, including Young's modulus ($E_{\parallel}$ and $E_{\perp}$), shear modulus ($G_{12}$ and $G_{13}$), bulk modulus ($K$), and Poisson's ratio ($\nu_{31}$ and $\nu_{12}$), are estimated from the five elastic stiffness constants according to Eq (3):

$$E_{33,\parallel} = C_{33} - \frac{C_{13}^2}{C_{11}-C_{66}} \quad (3a)$$

$$E_{11(22),\perp} = \frac{4C_{66}\left[C_{33}(C_{11}-C_{66}) - C_{13}^2\right]}{C_{11}C_{33} - C_{13}^2} \quad (3b)$$

$$G_{12} = C_{66} \quad (3c)$$

$$G_{13} = G_{23} = C_{44} \quad (3d)$$

$$K = \frac{-C_{13}^2 + C_{33}(C_{11}-C_{66})}{C_{11}-C_{66}-2C_{13}+C_{33}} \quad (3e)$$

$$\nu_{31} = \nu_{32} = \frac{C_{13}}{2(C_{11}-C_{66})} \quad (3f)$$

$$\nu_{12} = \nu_{21} = \frac{(C_{11}-2C_{66})C_{33} - C_{13}^2}{C_{11}C_{33} - C_{13}^2} \quad (3g)$$

$$\frac{\nu_{31}}{E_{33}} = \frac{\nu_{13}}{E_{11}} \quad (3h)$$

Access to elastic constants $C_{ij}$ allows the estimation of the effective elastic moduli of its polycrystalline structure based on the Voigt–Reuss–Hill approximation. The Voigt average, assuming constant strain field throughout the material, represents better the effective elastic properties of fiber-like structures in Eq. (4)[73].

$$G_{Voigt} = (C_{11} + C_{12} + 2C_{33} - 4C_{13} + 12C_{44} + 12C_{66})/30 \quad (4a)$$

$$K_{Voigt} = 2(C_{11} + C_{12} + 2C_{13} + C_{13}/2)/9 \quad (4b)$$

$$c_{L,Voigt} = \sqrt{\left(K_{Voigt} + \frac{4}{3}G_{Voigt}\right)/\rho} \quad (4c)$$

$$c_{T,Voigt} = \sqrt{G_{Voigt}/\rho} \quad (4d)$$

The computed longitudinal (transverse) sound velocities of polydomain LCE, $c_{L(T),Voigt} = 2460(1200)$ m/s, agree very well with the experimental values $c_{L(T),exp} = 2430(1110)$ m/s (dashed lines in Fig. 2b, Supplementary Fig. 4b) in favor of the performed analysis. To evaluate the optical anisotropy of LCE films, we combined transmission, reflection, and backscattering measurements to resolve dispersion relation $f(q)$ and hence the refractive index detailed in Supplementary Section 1.

## Micro-scale temperature wave analysis

The micro-scale temperature wave analysis approach calibrates the phase delay $\Delta\xi$ of a generated temperature wave with frequency $f$ crossing a short distance $d$. The heat flow could be treated as the one-dimensional type, thus the thermal diffusivity $\alpha$ could be estimated according to $\Delta\xi = -\sqrt{\frac{\pi f}{\alpha}}d - \frac{\pi}{4}$, where $d$ is usually the thickness of the cross-section of samples. To conduct the micro-scale temperature wave analysis measurements, the LCE samples were embedded in the epoxy resin (Mitsubishi Chemical Corporation., jER™ 828) with a thin metallic layer for Joule heating and a micro-

sized thermocouple type sensor sputtered at the front and rear surfaces of the cross-section, respectively. To realize the determination of two $\alpha$'s orthogonal to the director $\mathbf{n}$ ($\alpha_{\parallel}$ and $\alpha_{\perp}$), the LCEs with epoxy were sliced into two sets of which the directors were parallel and perpendicular to the cross-planes, respectively. A temperature wave was generated by the sinusoidal Joule heating, and its phase delay as crossing the sample was recorded by the lock-in amplifier. We measured several positions of LCEs for each sample and obtained the average $\alpha_{\parallel}$ and $\alpha_{\perp}$ for the aligned sample and $\alpha_r$ for polydomain sample.

## Data availability

All the data supporting the findings of this study are available within this article and its Supplementary Information files or are available from the corresponding author on reasonable request.

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

## Acknowledgements

Y.C. acknowledges the financial support by Shanghai Pujiang Program (Grant No. 20PJ1413800) and National Natural Science Foundation of China (Grant No. 12102304). Y.C. and G.F. acknowledge the financial support by ERC AdG SmartPhon (Grant No. 694977). J.M. acknowledges the financial support from a Japan KAKENHI grant (No. 20H04663), and a JST CREST grant (No. JPMJCR19I3). S.Y. acknowledges partial support by National Science Foundation (NSF) through the DMR/Polymer program, #DMR-2104841.

## Author contributions

G.F. and S.Y. conceived the ideas in the study; Y.C. and G.F. designed the research; J.L. synthesized liquid crystal elastomer films; Y.C., M.R., and B.G. collected the experimental data; Y.C., M.R., B.G., J.M., Y.S., and G.F. analyzed data; Y.C., J.L, J.M., S.Y., and G.F. wrote the manuscript. All authors discussed the results and commented on the manuscript.

## Funding

## Competing interests

The authors declare no competing interests.
