## [Peer Review File · Nature Communications]

On the origin of elasticity and heat conduction anisotropy of liquid crystal elastomers at gigahertz frequenciesREVIEWER COMMENTS

Reviewer #1 (Remarks to the Author):

The manuscript that presents experimental study of elastic anisotropy of liquid crystalline elastomers, specifically focussing on high-frequency regime. The work is interesting, but I found that the authors are apparently unaware of the large body of studies in this area, and this needs to be rectified. In the revision, they must review the earlier literature on the topics of their focus - and make it clear to the reader (and the referees) what is the novelty and the added value of this work. At present this is not clear, and I am not prepared to embark on my own investigation to establish this.

This comment essentially reflects the way the authors deal with their referencing. There are plenty of citations, but they do not serve the purpose.

As just one example: on page 2 of Introduction, the authors discuss the elastic anisotropy, and cite references 33-38... Of these, only refs. 34 (with Gleeson) and 38 (with Palffy-Muhoray) actually deal with elastic anisotropy at all - ref.33 is about polydomain-monodomain transition in LCE (another well-studied subject, but not one relevant to this paper), refs.35-36 are about monodomain LCE actuation, and ref.37 is about something else entirely. However, even ref.38, albeit necessary to mention, isn't the best citation: the question of elastic anisotropy in LCE was comprehensively studied in the work of Finkelmann, Greve and Warner (doi:10.1007/s101890170060), which the authors must study carefully.

The main premise of this work, that nobody studied LCE's at high frequency, is patently not true. Only very recently there was an article (doi: 10.1038/s41467-021-27012-1) where the Master Curves were shown up to very high frequencies (and earlier work of similar nature is cited).

The phonon propagation, the linear analysis similar to Eqs.1-4 here, and graphics similar to e.g. Fig.1(f), can be found in e.g. doi: 10.1098/rspa.2003.1153 and doi: 10.1103/PhysRevE.66.052701.

I think, the heat conduction (although not its anisotropy) was first studied in doi: 10.1140/epje/i2007-10266-4.

My point is that the authors "get away" with citing ref.1 (the LCE book) in many situations, but in reality there have been many more studies relevant to their topic that they need to find and study.

Reviewer #2 (Remarks to the Author):

The contribution from Cang and coauthors titled "On the origin of elasticity and heat conduction anisotropy of liquid crystal elastomers at gigahertz frequencies" presents an interesting body of work studying the anisotropic elastic properties of liquid crystalline elastomers when exposed to GHz frequencies such as those found in radio frequency packaging or 5G cellular networks. Due to the prevalence of technologies that utilize these frequencies, the relevance of the elastic response of materials to those conditions presents immediate relevance to a broad community.

There are a few points the authors should consider revising or clarifying prior to publication.

- In the introduction, paragraph 1, p. 2, the authors state: "Therefore, when heated above the nematic to isotropic phase transition temperature (TNI), LCEs will shrink in the aligned direction while expanding in the perpendicular direction with near zero stress^{7,8}. This phenomenon is often referred to as soft elasticity." Yakacki and coworkers recently reported (Nature Comm. (2021) 12:6677) on the well-established subject of soft elasticity, which describes a unique tensile (not thermal) LCE response: "Soft elasticity refers to the unique plateau-like tensile mechanical response of LCEs as described by theory pioneered by Warner and Terentjev..." This definition contrasts with the uncited one referenced by the authors and should be corrected and clarified at this point in the text, and elsewhere if it is applied as such.

Also in the introduction, paragraph 2, p 2, the authors state: "Fundamentally, the orientational contributions of the mesogens and the network polymer strands to the mechanical anisotropy can be frequency-dependent. However, for small molecule LC's, such as n-alkyl cyanobiphenyl (n-CB, n= 5-9)²² and phenylpyrimidine²³ LCs, the ratio of the longitudinal modulus in the parallel vs. orthogonal directions measured at GHz frequencies is typically $<1.15^{23,24}$ and hence much smaller than $E_{||}/E_{\perp}$ of LCEs from tensile testing." This may be a minor word choice issue, but it is not clear to me how these two statements relate to one another and this passage could use clarification.

In the introduction, paragraph 4, p. 3, the authors state, "Besides, the elastic response of LCEs exhibits a strong dependence on probed temperature and frequency due to its viscoelastic feature, consequently encumbering access to the purely linear elastic regime by conventional characterizations." It would be beneficial to the authors to defend this claim with 1 or more citations. Under static tensile testing conditions, a linear elastic regime is evident in many LCE, preceding onset of soft elastic plateau or strain hardening in the case of testing that is performed parallel to the LCE director. Admittedly, the presence of a linear elastic regime may be challenging at higher frequencies, but this should be detailed more in this section to clarify.

- While obviously great care was taken to prepare figures and text descriptions to aid in the visualization of the incident frequencies relative to the LCE polymer network, there are a few key aspects the authors could revise further in this section. First, the authors define the "incident angle β " within the caption of Fig. 1 but they introduce β within the main text without defining it until a subsequent paragraph. In the Fig. 1 caption there are other terms as well that are not as clearly defined in the main text discussion. It would help if the authors could duplicate some of the text that is currently only in the Figure caption to be present in the main text as well. In part d of Fig 1, the plane that the director "n" is occupying is not clear, and the planes through which the Q-L, Q-T, and P-T waves move through could also be clarified significantly through graphical revision.
- In Fig. 2b the authors describe q as normal to the x2-x3 plane, but the diagram makes it still look like it is in the x2-x3 plane as was the case in Fig 1. In Fig 2c, I only can identify 2 blue P-T points. Were there other P-T points collected? Are they just being overlapped by the Q-T points on that plot?
- In Fig 2d, plotting the Poisson's values here seemed out of place, especially since the other modulus types are described graphically as insets but the Poisson's are not. What does the arrow in Fig 2d signify? Could this be replotted to make the Poisson's values appear less of an after-thought?
- In the discussion from the end of page 9-page 10 there is some subset labeling using a slash and some using a parenthesis, and the subset numbers flip around. Is this distinction in notation intended and/or mathematically significant? More significantly in this section, the authors state, "while v_{31} ($=0.46 \pm 0.01$) is slightly lower compared to the v_{21} ($=0.52 \pm 0.03$)... Interestingly, the LCE becomes compressible ($v_{13}(23)+v_{12}(21) < 1$) at a normal strain to director n, violating the volume conservation assumption that is normally used for estimation of Poisson's ratio of LCEs..."

If we take the extremity of the stdev listed for the Poisson's measurements, in this case $0.47+0.55$, we calculate a value >1 , suggesting that the uncertainty for these measurements prevents the conclusion being made that the LCEs are compressible. The authors need to defend this claim with additional compelling data if they keep it in their discussion. The subsequent statement, "Compressibility was also found in the main-chain chiral nematic LCEs, both $v_{13}(23)$ and $v_{12}(21)$," would benefit from similar clarification.

- In the discussion on p 13, the authors state, "An alternative response is a sharp director rotation at a critical strain in analogy to the mechanical Fréedericksz transition (MFT) reported for few acrylate-based LCEs. ^{1,52,55,56,61} We are not aware of a tensile load curve displaying an MFT..."

The authors then proceed to describe how their data does comply with an MFT response. They state

that, "Fig. 3c, d, which both follow the local mesogenic orientation, comply with an MFT-like response. The latter deformation defines a stable ordering in this plateau-like region of which reorientation occurs sharply at a critical strain ϵ_c ..."

This becomes a point of confusion when comparing this description to Fig 3c, d, because there are no data points at or above the critical strain, ϵ_c , since the authors' materials fail below this threshold. Further, from the plotted data I cannot identify a point at which reorientation sharply occurs relative to the ϵ_c value, which seems necessary in order to make the claim that the response is MFT-like. It would be beneficial here for the authors to elaborate on how they were able to draw the conclusions they did, or to avoid generalizing the behavior to MFT-like.

- Relatedly, the authors state in the same discussion that, "Figure 3d implies a robust director rotation normal to the strain axis which is incompatible with SSE deformation." Could greater specificity be used to point to the data in Figure 3d that implies this? There is a lot going on in Fig 3d.

On line 15 the authors state that, "The opacity of both LCE films above TNI, precludes the record of q-dependent BLS spectra." This is an interesting feature of their materials. What do they suspect is leading to opacity of an isotropic LCE?

A few other minor grammatical points:

- Correct LC's and LCE's to LCs and LCEs throughout text when used as a plural.
- If it is possible to adjust the in-text references to specific sections of the Methods to something more succinct than "(ex. (Methods section 5 "Fabrication of Liquid Crystal Elastomer films")," the authors should consider this option because as-is it is distracting. Additional details would naturally be anticipated to be included in the Methods section.
- P.6 "being" the wavelength should be changed to "is"
- P. 10 "in an isothermal experiment but at sufficiently high frequencies." What constitutes sufficiently high frequencies? Sufficient to achieve what? Better worded as "Frequencies that are sufficiently high to..."
- P. 11 paragraph 2 refers to "the first BLS study", but this phrasing could be specified to help the reader picture what the authors are referring to.
- P12 paragraph 3, change potentials to potential
- P13 remove hyphen from stress-cost to low stress cost
- P15 change discontinued to discontinuous

If the authors consider the above corrections and do more to specify and clarify some of their key findings, this manuscript offers a solid and compelling analysis of LCE elastic response at GHz frequencies and could be resubmitted following substantial revision.

Reviewer #1 (Remarks to the Author):

The manuscript that presents experimental study of elastic anisotropy of liquid crystalline elastomers, specifically focussing on high-frequency regime. The work is interesting, but I found that the authors are apparently unaware of the large body of studies in this area, and this needs to be rectified. In the revision, they must review the earlier literature on the topics of their focus - and make it clear to the reader (and the referees) what is the novelty and the added value of this work. At present this is not clear, and I am not prepared to embark on my own investigation to establish this.

We thank Reviewer 1 for finding our “work interesting “and for the constructive criticisms on the cited literature.

This comment essentially reflects the way the authors deal with their referencing. There are plenty of citations, but they do not serve the purpose. As just one example: on page 2 of Introduction, the authors discuss the elastic anisotropy, and cite references 33-38... Of these, only refs. 34 (with Gleeson) and 38 (with Palffy-Muhoray) actually deal with elastic anisotropy at all - ref.33 is about polydomain-monodomain transition in LCE (another well-studied subject, but not one relevant to this paper), refs.35-36 are about monodomain LCE actuation, and ref.37 is about something else entirely. However, even ref.38, albeit necessary to mention, isn't the best citation: the question of elastic anisotropy in LCE was comprehensively studied in the work of Finkelmann, Greve and Warner (doi:10.1007/s101890170060), which the authors must study carefully.

Response: Triggered by this general criticism, we revisited related literature. Ref.33 is on the “polydomain-monodomain transition” as correctly pointed-out by the Reviewer. However, it reports the applicability of the time-temperature (t-T) superposition of the viscoelastic Young’s (E) modulus, leading to the elastic but isotropic modulus, $E \sim 2$ GPa at 20°C. Therefore, we think citation of this article is justified albeit in the wrong place. Ref. 35 (on thermoelasticity in kPa range), Ref.36 and Ref.37 are not relevant as correctly pointed out by the Reviewer. However, Ref.38 refers to the mechanical anisotropy of the five elastic constants at low frequencies and should be cited in clear context (4th paragraph of the Introduction).

We thank the Reviewer for pointing to the missing reference [“The elastic anisotropy of nematic elastomers” by H. Finkelmann, A. Greve and M. Warner, Eur. J. Phys. E., 5, 281 (2001)] which is indeed relevant as it reports the shear modulus (G) of a monodomain LCE parallel ($G_{\text{para}} \sim 90$ kPa) and normal ($G_{\text{perp}} \sim 40$ kPa) to director orientation, when its rotation is prohibited. They are now cited as ref. 32. However, as we pointed out in our manuscript, only two orthogonal (symmetry) directions presented in general LCE literature at the low frequency do not allow access to the complete (five independent elastic constants) anisotropy of the literally elastic tensor, as in the present case. Further, G_{para} and G_{perp} are not the elastic components falling in the GPa range (Fig. 2d) but interestingly $G_{\text{para}}/G_{\text{perp}} \approx E_{\text{para}}/E_{\text{perp}} \sim 2.2$ is comparable to the elastic Young’s modulus anisotropy (Fig. 2d). Note that complete determination of the elasticity by BLS is obtained at zero strain!

Changes: Introduction, 4th paragraph: The original refs 35-37 are deleted, Ref. 33 is now 34 and the added new reference (EPJ E 2001) is ref. 32.

Anisotropic elasticity, 3rd paragraph: “...for monodomain LCE.¹⁴ Even much lower \$G_{\text{l(L),tensile}} \approx 90(40)\$ kPa was reported for a monodomain LCE, when the director rotation under stress was prohibited³². So the anisotropic elasticity at low frequencies in the rubbery state assumes low moduli values^{32, 53, 54}.

Elasticity response to external stimuli, 1st paragraph: an MFT, which is so far revealed through tracking of the director rotation in LCEs by optomechanical and scattering techniques throughout tensile testing.^{32,59,60}

The main premise of this work, that nobody studied LCE's at high frequency, is patently not true. Only very recently there was an article (doi: 10.1038/s41467-021-27012-1) where the Master Curves were shown up to very high frequencies (and earlier work of similar nature is cited).

Response: We thank the Reviewer for pointing out this recent *Nat. Commun.* paper (2021). Nevertheless, it does not impact the novelty statement of our work as we commented on a reference of similar nature (an earlier paper (2016, now ref. 34)) when we discussed in the subsection of Anisotropic elasticity, end of the 3rd paragraph: “In fact, the polymer network strand dynamics controlling the LCE viscoelasticity fall in the intermediate MHz frequency range⁵², and the large discrepancy in the values of E clearly reflects its strong frequency dependence for the viscoelastic LCE.³³” We note the frequencies studied in the *Nat. Commun.* (2021) is in range of 0.01–200 Hz, whereas the high frequency we are talking about in our manuscript is GHz. In addition, the high frequency storage E' values reported in the *Nat. Commun.* (2021) are isotropic and bear large error bars in the t-T superposition curves. More importantly, the assumption of a robust structure over the broad scanned T-range is questionable. As mentioned in our original manuscript (end of the 3rd paragraph) the high frequency E should be insensitive to crosslinking density, which is in sharp contrast to Fig. 2a of *Nat. Commun.* (2021) (about 1 GPa and 3 GPa respectively for the low and high crosslinking density LCE, respectively). Instead, the dynamic mechanical temperature scan at low frequencies yields the elastic E of the frozen (elastic) LCEs at very low temperatures (Fig. 1b).

Changes: An additional comment is added in subsection Anisotropic elasticity, end of the 3rd paragraph: In this context, the recently reported storage E' from t-T superposition of dynamic frequency scan viscoelastic tests displays a crosslinking dependence (from 1 to about 3 GPa over the real frequency range of 0.01–200 Hz).³⁵ While this isotropic value is comparable to E_{\perp} in Fig. 2d, the large error of the t-T superposition and more importantly the assumption of a robust structure over the broad scanned T-range renders the reported strong crosslinking dependence questionable.³⁵ Notably, the t-T superposition applies for either nematic or isotropic states but does not hold across the phase transitions.³⁴⁻³⁷

The phonon propagation, the linear analysis similar to Eqs.1-4 here, and graphics similar to e.g. Fig.1(f), can be found in e.g. doi: 10.1098/rspa.2003.1153 and doi: 10.1103/PhysRevE.66.052701.

Response: We thank the Reviewer for the two additional theoretical references on the propagation of shear waves LCE. a) Propagation of acoustic wave in nematic elastomers by E. M. Terentjev, et al., *Phys. Rev. E* (2002): The propagation is dissipative due to the proximity of the shear wave frequencies to the inverse of the Rouse time of the network resulting to strong attenuation. The similarity of the polar plot (Fig. 2) with our Fig. 1f is apparent because of the different wave polarization (shear vs longitudinal), attenuation (dissipative vs elastic response) due to the vastly different frequencies in the two cases. b) Low frequency acoustic waves in nematic elastomers by L. J. Franklin, et al., *Proc. R. Soc. Lond.* (2003). As per title, this is similar theory dealing with dissipative low frequency shear waves. Note that the Christoffel's equation is the solution of wave equation in the continuum mechanics elasticity. To contrast with our case, we cite only a).

Changes: Anisotropic Elasticity, 2nd paragraph:Table S1. Note the absence of dissipation in the Christoffel equation, in contrast to elastic wave propagation at low frequencies comparable to the network Rouse rate.^{36,45}

I think, the heat conduction (although not its anisotropy) was first studied in doi: 10.1140/epje/i2007-10266-4.

Response: We thank the Reviewer for the suggested reference, K. K. Hon, D. Corbett and E. M. Terentjev, Eur. Phys. J. E, 25, 83 (2008). However, this paper does not deal with experimental heat conductivity but modelling of the bending kinetics in LCE cantilever (Fig. 5,6) using the thermal diffusion as an adjustable parameter. As clearly stated by the authors of the paper at the end of p.88 before Conclusions “We are not aware of any measurements of thermal diffusion in nematic LCE”. Nevertheless, the adjustable value estimated as $\sim 1.5 \cdot 10^{-7} \text{ m}^2/\text{s}$ is supported by our experimental $\alpha_{\text{random}} = (1.20 \pm 0.03) \cdot 10^{-7} \text{ m}^2/\text{s}$ as shown in Fig. 4d. The refs. 28 and 30 cited in our original manuscript are all experimental ones. However, based on the content of the recommended paper, our anisotropic heat conductivity can impact applications of thermally stimulated LCE.

Changes: Discussion, 3rd paragraph: *In analogy to the photomechanical response of LCEs^{71,72} that leads to bending, inhomogeneous deformations can benefit from non-uniform heating in the absence of dyes harnessing the anisotropic thermal conduction in LCEs.*

My point is that the authors "get away" with citing ref.1 (the LCE book) in many situations, but in reality there have been many more studies relevant to their topic that they need to find and study.

Response: Ref.1 is a book on LCE with two tutorial chapters (6 and 7), which we consider useful for curious readers in the field. Nevertheless, we have now cited additional original literature relevant to our topic. Specifically, reference that show the novelty of the study, truly elastic anisotropy from a non-destructive, zero strain optical technique.

Changes: *New references:*

[22] Mistry, D. *et al.* Soft elasticity optimises dissipation in 3D-printed liquid crystal elastomers. *Nature Communications* **12**, 6677 (2021).

[32] Finkelmann, H., Greve, A. & Warner, M. The elastic anisotropy of nematic elastomers. *The European Physical Journal E* **5**, 281-293 (2001).

[36] Zanna, J., Stein, P., Marty, J., Mauzac, M. & Martinoty, P. Influence of molecular parameters on the elastic and viscoelastic properties of side-chain liquid crystalline elastomers. *Macromolecules* **35**, 5459-5465 (2002).

[37] Giamberini, M., Ambrogio, V., Cerruti, P. & Carfagna, C. Viscoelasticity of main chain liquid crystalline elastomers. *Polymer* **47**, 4490-4496 (2006).

[45] Terentjev, E. M., Kamotski, I. V., Zakharov, D. D. & Fradkin, L. J. Propagation of acoustic waves in nematic elastomers. *Physical Review E* **66**, 052701 (2002).

[53] Rogez, D., Francius, G., Finkelmann, H. & Martinoty, P. Shear mechanical anisotropy of side chain liquid-crystal elastomers: influence of sample preparation. *The European physical journal. E, Soft matter* **20**, 369-378 (2006).

[61] Clarke, S. M., Hotta, A., Tajbakhsh, A. R. & Terentjev, E. M. Effect of crosslinker geometry on equilibrium thermal and mechanical properties of nematic elastomers. *Physical review. E, Statistical, nonlinear, and soft matter physics* **64**, 061702 (2001).

[71] Yu, Y., Nakano, M. & Ikeda, T. Directed bending of a polymer film by light. *Nature* **425**, 145-145 (2003).

[72] Camacho-Lopez, M., Finkelmann, H., Palffy-Muhoray, P. & Shelley, M. Fast liquid-crystal elastomer swims into the dark. *Nature Materials* **3**, 307-310 (2004).

Reviewer #2 (Remarks to the Author):

The contribution from Cang and coauthors titled “On the origin of elasticity and heat conduction anisotropy of liquid crystal elastomers at gigahertz frequencies” presents an interesting body of work studying the anisotropic elastic properties of liquid crystalline elastomers when exposed to GHz frequencies such as those found in radio frequency packaging or 5G cellular networks. Due to the prevalence of technologies that utilize these frequencies, the relevance of the elastic response of materials to those conditions presents immediate relevance to a broad community. There are a few points the authors should consider revising or clarifying prior to publication.

We thank this Reviewer for finding our contribution interesting and in particular for all twelve raised points which helped us to further strengthen certain statements.

*1. In the introduction, paragraph 1, p. 2, the authors state: “Therefore, when heated above the nematic to isotropic phase transition temperature (TNI), LCEs will shrink in the aligned direction while expanding in the perpendicular direction with near zero stress^{7,8}. This phenomenon is often referred to as soft elasticity.” Yakacki and coworkers recently reported (*Nature Comm.* (2021) 12:6677) on the well-established subject of soft elasticity, which describes a unique tensile (not thermal) LCE response: “Soft elasticity refers to the unique plateau-like tensile mechanical response of LCEs as described by theory pioneered by Warner and Terentjev...” This definition contrasts with the uncited one referenced by the authors and should be corrected and clarified at this point in the text, and elsewhere if it is applied as such.*

Response: We thank the Reviewer for pointing out this apparently incorrect statement. In fact, semi-soft elasticity is correctly introduced in the Elasticity response to external stimuli.

Changes: Introduction: the statement “~~This phenomenon is often referred as soft elasticity.~~” is now deleted. Elasticity response to external stimuli: 1st paragraph: resembling liquids. The rotation of the network strand shape proceeds with no space distortion and energy cost and ideally LCEs enable absorption of strain energy at constant stress.²²

2. Also in the introduction, paragraph 2, p 2, the authors state: “Fundamentally, the orientational contributions of the mesogens and the network polymer strands to the mechanical anisotropy can be frequency-dependent. However, for small molecule LC’s, such as n-alkyl cyanobiphenyl (n-CB, n= 5-9)²² and phenylpyrimidine²³ LCs, the ratio of the longitudinal modulus in the parallel vs. orthogonal directions measured at GHz frequencies is typically <1.15^{23,24} and hence much smaller than E_{\parallel}/E_{\perp} of LCEs from tensile testing.” This may be a minor word choice issue, but it is not clear to me how these two statements relate to one another and this passage could use clarification.

Response: Indeed, this sentence could be misunderstood. We have clarified it in the revised Ms.

Change: Introduction, 2nd paragraph: “...can be frequency-dependent due to their very different dynamics. For small molecule LCs, such as n-alkyl cyanobiphenyl (n-CB, n=5-9)²⁰ and phenylpyrimidine²¹ LCs, the

ratio of the longitudinal modulus in the parallel vs. orthogonal directions measured and GHz frequencies is typically $<1.15^{21}$; in the absence of a polymer network, this anisotropy reflects contributions solely by the mesogens. For LCE, E_{\parallel}/E_{\perp} from low frequency tensile testing is much larger (~ 3.3)²² and is considered as the static anisotropy value due to the contribution from both the aligned network and the mesogens.

3. In the introduction, paragraph 4, p. 3, the authors state, “Besides, the elastic response of LCEs exhibits a strong dependence on probed temperature and frequency due to its viscoelastic feature, consequently encumbering access to the purely linear elastic regime by conventional characterizations.” It would be beneficial to the authors to defend this claim with 1 or more citations. Under static tensile testing conditions, a linear elastic regime is evident in many LCE, preceding onset of soft elastic plateau or strain hardening in the case of testing that is performed parallel to the LCE director. Admittedly, the presence of a linear elastic regime may be challenging at higher frequencies, but this should be detailed more in this section to clarify.

Response: We thank this Reviewer for raising this pertinent point that is often confused in the published literature. Elastic response is warranted either by dynamic mechanical analysis at low scan frequency in the glassy (frozen) material (Fig. 1b of ref. 4) or at sufficiently high frequency in the rubbery state (above T_g) (by BLS in the present work). In either case, E falls in the GPa (and not in MPa) range. It is correct that “under static tensile testing conditions, a linear elastic regime is evident in many LCE” in the rubbery state but E is either few MPa (14, ref. 4,53) or even tenths of kPa (33) as both mesogens orientation and chain conformation dynamics are not arrested.

Change: ‘Besides, the **mechanical elastic** response of LCE exhibits a strong dependence on probed temperature and frequency due to its viscoelastic feature³⁴⁻³⁷, consequently encumbering access to the purely **linear** elastic $E_{\parallel,\perp}$ **regime** of LCE (in the rubbery state) by conventional characterization. **Elastic material response ($E_{\parallel,\perp}$ typically in the GPa range) is ensured for sufficiently high frequency (compared to the fastest material dynamics) applied isothermally and not artificially by time-temperature (t-T) superposition of low frequency scan tests at different temperatures.**³⁴⁻³⁷ Therefore, there remains ...

4. While obviously great care was taken to prepare figures and text descriptions to aid in the visualization of the incident frequencies relative to the LCE polymer network, there are a few key aspects the authors could revise further in this section.

First, the authors define the “incident angle β ” within the caption of Fig. 1 but they introduce β within the main text without defining it until a subsequent paragraph. In the Fig. 1 caption there are other terms as well that are not as clearly defined in the main text discussion. It would help if the authors could duplicate some of the text that is currently only in the Figure caption to be present in the main text as well. In part d of Fig 1, the plane that the director “n” is occupying is not clear, and the planes through which the Q-L, Q-T, and P-T waves move through could also be clarified significantly through graphical revision.

Response: We thank the reviewer for the careful observation. Accordingly, we now define the incident angle β and wave vector q and modified Fig.1d in the revised manuscript.

Changes: Anisotropic elasticity, 1st paragraph: ...The sound velocities can be directly obtained from the frequency f of the acoustic phonons in the BLS spectra recorded at **different magnitudes-of-the** wave vector $q = k_s - k_i$ **selected through** the orientation angle, ϕ , relatively to the director \mathbf{n} of the monodomain LCE film (Fig. 1b, c)⁴⁴, where k_i and k_s are the wave vectors of the incident and scattered light, respectively, and q denotes the magnitude of q . Molecular and polymer...(Fig.1b). In the transmission scattering geometry

phonons propagate parallel to the (x_2, x_3) film plane with the wavenumber $q = (4\pi/\lambda) \sin\beta$, where $\lambda (=532\text{nm})$ is the wavelength of the laser light in vacuum and β is the incident angle (Fig. 1b).

Fig. 1d is now modified.

5. (a) In Fig. 2b the authors describe q as normal to the x_2 - x_3 plane, but the diagram makes it still look like it is in the x_2 - x_3 plane as was the case in Fig 1.

Response: The Reviewer is right with the wrong impression from the two-dimensional inset to Fig. 2b.

Change: The schematic backscattering geometry in Fig. 2b is now improved with the inset in the 3-D form.

5.(b) In Fig 2c, I only can identify 2 blue P-T points. Were there other P-T points collected? Are they just being overlapped by the Q-T points on that plot?

Response: Indeed, we have collected two P-T (blue points) data at high orientation angle ϕ 's in Fig. 2c.

Change: Supplementary Information Fig. S5: The two P-T data at high orientation angle ϕ 's (Fig. 2c) allows for determination of C_{66} according to $\rho \left(\frac{2\pi}{q}\right)^2 f_{P-T}^2 = \sin^2 \phi C_{66} + \cos^2 \phi C_{44}$ (solid line in Fig. 2c). $f_{P-T} = f_{Q-T}$ is valid when $\phi=0^\circ$. Other four independent elastic constants (C_{11} , C_{33} , C_{13} and C_{44}) are mainly determined by $f_{Q-L}(\phi)$ and $f_{Q-T}(\phi)$.

6. In Fig 2d, plotting the Poisson's values here seemed out of place, especially since the other modulus types are described graphically as insets but the Poisson's are not. What does the arrow in Fig 2d signify? Could this be replotted to make the Poisson's values appear less of an after-thought?

Response: We would prefer to show the Poisson's ratio along with the engineering moduli in the same plot. Albeit no graphical illustration is presented in Fig.2d, the meaning of the two coefficients is given in p.10.

Changes: Figure 2 caption: (d). ...(shaded columns). The black arrow points to the right axis of the Poisson's ratio, ν . (f). The viscosity coefficients, $\eta_{\parallel}=\eta_{33}$ along and $\eta_{\perp}=\eta_{11}$ normal to the director of monodomain, and η_r of polydomain LCE, ordinary and extraordinary refractive index (n_o and n_e) of monodomain LCE, and refractive index n_r of polydomain LCE. ~~are shown in the left and right panels, respectively.~~ The black arrows point to the right viscosity and to the left refractive index in the left and right axis, respectively.

7. (a) In the discussion from the end of page 9-page 10 there is some subset labeling using a slash and some using a parenthesis, and the subset numbers flip around. Is this distinction in notation intended and/or mathematically significant?

Response: The Reviewer is right with the unclear but distinct notation. The slash and parenthesis have different meaning. For example, $G_{13/23}$ and $\nu_{31/32}$, denote the values with each subset labeling separated by the slash are equivalent, e.g., $G_{13}=G_{23}$, and $\nu_{31}=\nu_{32}$. In contrast, the parenthesis, such as $\nu_{13(23)+\nu_{12(21)} < 1$ and $E_{l(\perp),\text{tensile}} =10(3)$ MPa, suggest the corresponding relation of values either inside or outside the parenthesis, e.g., $\nu_{13}+\nu_{12} < 1$ and $\nu_{23}+\nu_{21} < 1$, and $E_{l,\text{tensile}} =10$ MPa and $E_{\perp,\text{tensile}} =3$ MPa.

Change: To avoid this complication, we remove the slash notation: p.10...sliding $G_{13}=G_{23}$...shear anisotropy $G_{12} \gg G_{13} \dots$ Both $\nu_{31}=\nu_{32}$ and $\nu_{2,1}$. p.10: ...nematic LCE both $\nu_{13}=\nu_{32}$ and $\nu_{12}=\nu_{21}$

8. (b) More significantly in this section, the authors state, “while ν_{31} ($=0.46 \pm 0.01$) is slightly lower compared to the ν_{21} ($=0.52 \pm 0.03$)... Interestingly, the LCE becomes compressible ($\nu_{13}(23)+\nu_{12}(21) < 1$) at a normal strain to director \mathbf{n} , violating the volume conservation assumption that is normally used for estimation of Poisson’s ratio of LCEs...” If we take the extremity of the stdev listed for the Poisson’s measurements, in this case $0.47+0.55$, we calculate a value >1 , suggesting that the uncertainty for these measurements prevents the conclusion being made that the LCEs are compressible. The authors need to defend this claim with additional compelling data if they keep it in their discussion. The subsequent statement, “Compressibility was also found in the main-chain chiral nematic LCEs, both $\nu_{13}(23)$ and $\nu_{12}(21)$,” would benefit from similar clarification.

Response: We agree with the Reviewer that the concluded compressibility of LCE could be doubtful given the uncertainty of the measured Poisson’s ratio. However, the unconstrained Poisson’s ratio (beyond the range of 0 to 0.5 allowed for isotropic elastomers) in the anisotropic monodomain LCEs allows for the presence of compressibility in the designed chiral nematic LCE.¹⁷

Change: “Interestingly, the LCE may be compressible ($\nu_{13}(23)+\nu_{12}(21) < 1$) at a normal strain to director \mathbf{n} , violating the volume conservation assumption that is normally used for estimation of Poisson’s ratio of any rubbery materials^{38,50}. The Poisson’s ratio of anisotropic LCE should have no boundaries⁵² unlike the isotropic elastomers ($0 < \nu \leq 0.5$). In fact, compressibility was found in main-chain chiral nematic LCE, where both ν_{13} and ν_{12} were found larger than 0.5 allowing for a significant volume change as applying a small transverse deformation.¹⁵”

9. In the discussion on p 13, the authors state, “An alternative response is a sharp director rotation at a critical strain in analogy to the mechanical Fréedericksz transition (MFT) reported for few acrylate-based LCEs. 1,52,55,56,61 We are not aware of a tensile load curve displaying an MFT...” The authors then proceed to describe how their data does comply with an MFT response. They state that, “Fig. 3c, d, which both follow the local mesogenic orientation, comply with an MFT-like response. The latter deformation defines a stable ordering in this plateau-like region of which reorientation occurs sharply at a critical strain s_c ...” This becomes a point of confusion when comparing this description to Fig 3c, d, because there are no data points at or above the critical strain, s_c , since the authors’ materials fail below this threshold. Further, from the plotted data I cannot identify a point at which reorientation sharply occurs relative to the s_c value, which seems necessary in order to make the claim that the response is MFT-like. It would be beneficial here for the authors to elaborate on how they were able to draw the conclusions they did, or to avoid generalizing the behavior to MFT-like.

Response: The Reviewer has correctly pointed out that our LCE breaks at s_f just before the estimated critical strain s_c for an abrupt MFT-like director rotation. Fig. 3c,d show that straining LCE up to s_f both sound velocities and linewidths are virtually constant implying robust director orientation. In the case of SSE deformation, director rotation should have already started above $s > 0.25$ based on the tensile load curve of our LCE sample. Hence albeit LCE breaks before s_c , the constant sound velocity (Fig. 3c) and attenuation (Fig. 3d), in particular for the perpendicular stretching ($\mathbf{u} \perp \mathbf{n}$), apparently contradicts a continuous director rotation between 0.25-0.5 implied by an SSE deformation while conforms to an MFT behavior. As shown in the inset of Fig.4b, c_l drops significantly when the director orientation changes.

Change: While for $\mathbf{u} \parallel \mathbf{n}$ stretching ..., the experimental observation of a virtually constant anisotropy (c_l/c_t) in Fig. 3c appears counterintuitive as the director field would have been continuously reoriented toward the

direction of the stress and hence manifested in a decrease of c_{\parallel}/c_{\perp} . The corresponding the robust hypersonic sound velocity anisotropy c_{\parallel}/c_{\perp} upon perpendicular ($\mathbf{u}\perp\mathbf{n}$) stretching in Fig. 3c conforms to an MFT-like response with constant director orientation normal to the direction of stress; for an SSE deformation of our LCE sample, director rotation should have already started above $s > 0.25$. Instead, MFT-like deformation defines a stable....

10. Relatedly, the authors state in the same discussion that, "Figure 3d implies a robust director rotation normal to the strain axis which is incompatible with SSE deformation." Could greater specificity be used to point to the data in Figure 3d that implies this? There is a lot going on in Fig 3d.

Response: We agree that this short sentence was dense and is now expanded.

Change: Albeit that our LCE breaks at s_f just before the estimated critical strain s_c for an abrupt director rotation towards the stretching direction, the MFT-like response is further supported by the Q-L phonon linewidths plotted vs. the strain in Fig. 3d. The phonon attenuation anisotropy ($\Gamma_{\parallel}/\Gamma_{\perp}$) remains, like the sound velocity anisotropy, within the experimental errors throughout the stretching process. Since the linewidth anisotropy, $\Gamma_{\parallel}/\Gamma_{\perp} = \eta_{33}/\eta_{11}$, the phonon attenuation anisotropy also follows the local mesogenic orientation and hence Figure 3d implies a robust director rotation normal to the strain axis ($\mathbf{u}\perp\mathbf{n}$) which again is compatible for an MFT-like response. This notion....

11. On line 15 the authors state that, "The opacity of both LCE films above T_{NI} , precludes the record of q -dependent BLS spectra." This is an interesting feature of their materials. What do they suspect is leading to opacity of an isotropic LCE?

Response: The appearance of opacity after the nematic-isotropic phase transition could be induced by the multiple scattering from the microscopic domains which interestingly become discernible in the isotropic state (Fig. 3 in [Xia Yu, Xinyue Zhang, and Shu Yang. "Instant Locking of Molecular Ordering in Liquid Crystal Elastomers by Oxygen-Mediated Thiol–Acrylate Click Reactions." *Angewandte Chemie* 130, 5767-5770 (2018)]). We found the LCE with the same mesogen (RM82) on the main chain in ref [Ware, T. H., and T. J. White. "Programmed liquid crystal elastomers with tunable actuation strain." *Polymer Chemistry* 6. 4835-4844 (2015).] also turned opaque at $T > T_{ni}$. However, what parameter whether the chemical structures, the synthesis method, the chain length of the oligomers, or the composition of the precursor plays a role remains still unclear.

Changes: The opacity of both LCE films above T_{NI} , which may result from the multiple scattering from the discernible microscopic domains⁴⁰, precludes the recording of q -dependent BLS spectra.

12. A few other minor grammatical points:

We indeed thank the Reviewer for the very careful proofreading of our original Ms. We have corrected these words/sentences in the revised manuscript as described below:

- Correct LC's and LCE's to LCs and LCEs throughout text when used as a plural.

We use LCs and LCEs now as suggested.

- *If it is possible to adjust the in-text references to specific sections of the Methods to something more succinct than “(ex. (Methods section 5 “Fabrication of Liquid Crystal Elastomer films”),” the authors should consider this option because as-is it is distracting. Additional details would naturally be anticipated to be included in the Methods section.*

Change is made as suggested.

- P.6 “being” the wavelength should be changed to “is”

Change is made as suggested. • P. 10 “in an isothermal experiment but at sufficiently high frequencies.” What constitutes sufficiently high frequencies? Sufficient to achieve what? Better worded as “Frequencies that are sufficiently high to....”

but at ~~sufficiently high~~ frequencies that are sufficiently high to exclude the network contribution.

- P. 11 paragraph 2 refers to “the first BLS study”, but this phrasing could be specified to help the reader picture what the authors are referring to.

from the ~~first~~ present BLS study (Fig. 2d).

- P12 paragraph 3, change potentials to potential

Change is made as suggested.

- P13 remove hyphen from stress-cost to low stress cost

Change is made as suggested.

- P15 change discontinued to discontinuous

Change is made as suggested.

If the authors consider the above corrections and do more to specify and clarify some of their key findings, this manuscript offers a solid and compelling analysis of LCE elastic response at GHz frequencies and could be resubmitted following substantial revision.

We do hope that we have convincingly addressed all raised points that indeed helped to clarify the important messages of our study.

REVIEWERS' COMMENTS

Reviewer #1 (Remarks to the Author):

The authors have addressed the points that concerned me, namely, placing their work in a better context of the earlier literature. I find their responses to the other Referee report also adequate.

I am still concerned by a possible conceptual misunderstanding here, as the term "elastomer" (whether liquid-crystallin or ordinary) is not really meaningful at frequencies above the dynamic glass transition, which appears to be around 10-100kHz. So at the frequencies of interest (MHz to GHz), this is an anisotropic glass. So none of the theories or ideas developed for liquid crystalline elastomers (involving incompressibility, a vast discrepancy between shear and compression waves, only two independent parameters defining the elastic response, etc) are applicable. The authors are working with what is known as a 'transversely isotropic solid', with indeed 5 independent elastic constants, or multiple Poisson ratios. What they do with this is correct and valid - whether it is of any interest to the readers remains to be seen.

So I recommend publication.

Reviewer #2 (Remarks to the Author):

The authors have adequately responded to my initial queries.

Reviewer #1 (Remarks to the Author):

The authors have addressed the points that concerned me, namely, placing their work in a better context of the earlier literature. I find their responses to the other Referee report also adequate.

I am still concerned by a possible conceptual misunderstanding here, as the term "elastomer" (whether liquid-crystallin or ordinary) is not really meaningful at frequencies above the dynamic glass transition, which appears to be around 10-100kHz. So at the frequencies of interest (MHz to GHz), this is an anisotropic glass. So none of the theories or ideas developed for liquid crystalline elastomers (involving incompressibility, a vast discrepancy between shear and compression waves, only two independent parameters defining the elastic response, etc) are applicable. The authors are working with what is known as a 'transversely isotropic solid', with indeed 5 independent elastic constants, or multiple Poisson ratios. What they do with this is correct and valid - whether it is of any interest to the readers remains to be seen.

The Reviewer's nomenclature concern is based on the different responses of liquid crystal elastomer (LCE) on frequency. However, the material's nomenclature is usually defined by its chemical nature and common wisdom in literature. For example, polymers can have different physical states, glassy or rubbery depending on temperatures or frequency. It is well known that the frequency-temperature defines the two limits but the nomenclature does not change because of changed behaviours at a given temperature or frequency. Otherwise, it will cause a lot of confusion to readers. Elastomer is about polymer network that is lightly crosslinked and has a glass transition below room temperature, so it appears rubbery at room temperature. LCE is a special kind of elastomer that has mesogenic units in the network, which is elastomeric or rubbery at room temperature. Already, our title clearly states the elastic nature of LCEs at high (GHz) frequencies. Further in the Introduction (at the end of the 2nd paragraph) of our manuscript, we clarified: "When probed at low frequencies, the LCE is in the elastomeric state where the network polymer strands are mobile, whereas they become frozen when probed at GHz," Therefore, we prefer not to make any new change, as our material fits the definition of LCE.

So I recommend publication.

We thank Reviewer for his/her recommendation.